# Coursing hyenas and stalking lions: The potential for inter- and intraspecific interactions

Nancy A. Barker[1]*, Francois G. Joubert[2], Marthin Kasaona[3], Gabriel Shatumbu[3],
Vincent Stowbunenko[4], Kathleen A. Alexander[5], Rob Slotow[6‡], Wayne M. Getz[7,8‡]

1 School of Life Sciences, University of KwaZulu-Natal, Durban, South Africa, 2 African Wildlife Veterinarian, Outjo, Namibia, 3 Etosha Ecological Institute, Ministry of Environment and Tourism, Okaukuejo, Namibia, 4 Department of Computer Science, San José State University, San Jose, California, United States of America, 5 Department of Fish and Wildlife Conservation, Virginia Tech, Blacksburg, Virginia, United States of America, 6 Oppenheimer Fellow in Functional Ecology, Centre for Functional Ecology, School of Life Sciences, University of KwaZulu-Natal, Pietermaritzburg, South Africa, 7 Department of Environmental Science, Policy & Management, University of California, Berkeley, California, United States of America, 8 School of Mathematical Sciences, University of KwaZulu-Natal, Durban, South Africa

☯ These authors contributed equally to this work.
‡ RS and WMG also contributed equally to this work.
* nancybarkerzoologist@gmail.com

**Data Availability Statement:** https://doi.org/10.5281/zenodo.5949231.

**Funding:** NAB. PGSD3-404001-2011. Natural Sciences and Engineering Research Council of

## Abstract

Resource partitioning promotes coexistence among guild members, and carnivores reduce interference competition through behavioral mechanisms that promote spatio-temporal separation. We analyzed sympatric lion and spotted hyena movements and activity patterns to ascertain the mechanisms facilitating their coexistence within semi-arid and wetland ecosystems. We identified recurrent high-use (revisitation) and extended stay (duration) areas within home ranges, as well as correlated movement-derived measures of inter- and intra-specific interactions with environmental variables. Spatial overlaps among lions and hyenas expanded during the wet season, and occurred at edges of home ranges, around water-points, along pathways between patches of high-use areas. Lions shared more of their home ranges with spotted hyenas in arid ecosystems, but shared more of their ranges with conspecifics in mesic environments. Despite shared space use, we found evidence for subtle temporal differences in the nocturnal movement and activity patterns between the two predators, suggesting a fine localized-scale avoidance strategy. Revisitation frequency and duration within home ranges were influenced by interspecific interactions, after land cover categories and diel cycles. Intraspecific interactions were also important for lions and, important for hyenas were moon illumination and ungulates attracted to former anthrax carcass sites in Etosha, with distance to water in Chobe/Linyanti. Recursion and duration according to locales of competitor probabilities were similar among female lions and both sexes of hyenas, but different for male lions. Our results suggest that lions and spotted hyenas mediate the potential for interference competition through subtle differences in temporal activity, fine-scale habitat use differentiation, and localized reactive-avoidance behaviors. These findings enhance our understanding of the potential effects of interspecific interactions among large carnivore space-use patterns within an apex predator system and show

Canada (NSERC). https://www.nserc-crsng.gc.ca/
Students-Etudiants/PG-CS/BellandPostgrad-
BelletSuperieures_eng.asp. WMG. GM83863.
National Institutes of Health (NIH) Grant. https://
www.nih.gov/grants-funding RS. University of
KwaZulu-Natal funding. https://lifesciences.ukzn.
ac.za/ KAA. NSF CNH2: 2009717, NSF Expeditions:
1918770 The funders had no role in study design,
data collection and analysis, decision to publish, or
preparation of the manuscript.

**Competing interests:** The authors have declared
that no competing interests exist.

adaptability across heterogeneous and homogeneous environments. Future conservation plans should emphasize the importance of inter- and intraspecific competition within large carnivore communities, particularly moderating such effects within increasingly fragmented landscapes.

## Introduction

Spatio-temporal partitioning helps stabilize multi-species communities in which more than one species use the same resource [1–3]. More specifically, species whose ranges overlap forage different types of food either preferentially or opportunistically, feed at different temporal schedules [4–10], demonstrate habitat separation, exhibit nonsynchronous spatial overlap or temporal partitioning [11–16]. Environmental heterogeneity provides temporary refugia where the risk of competition and injury is reduced [15, 17]. In addition, when there is an abundance of surplus resources the amount of food attracts numerous competitors, such that the energy required to exclude them becomes costly, and, therefore, competition ceases [18]. Resource use varies widely among sympatric carnivores [12, 16, 19–25], including African carnivores [3, 8, 26–33], and is presumed to promote coexistence [34–37].

Lions (*Panthera leo*) and spotted hyenas (*Crocuta crocuta*) are mainly crepuscular and nocturnal predators that demonstrate at least an 80% overlap between their daily activity budgets [14]. Although both species are sometimes active during cool winter days [38–40], they do not appear to use temporal partitioning to avoid interference competition [41]. The population densities of both lions and spotted hyenas (hereafter hyena) are primarily influenced by the abundance of prey, and are, thus, positively correlated in some areas [42, 43]. As increasing prey abundance leads to an increase in the densities of both predators, however, the potential for interference competition increases with the likelihood of interspecific encounters [8]. Nonetheless, it appears that hyenas derive benefits from sharing areas with lions. Hyenas appropriated up to 100% of lion kills in the Ngorongoro Crater when adult male lions were absent [44]. In the Amboseli National Park, hyenas did not avoid lion sounds from audio-call stations, and sometimes even approached these stations in response to lion roars [45]. This type of behavior likely persists because avoiding lions may cost hyenas missed scavenging opportunities, given the high degree of overlap in their diets [10].

The ecological dynamics between lions and hyenas are thus complex, and coexistence may be occurring due to the spatiotemporal partitioning of resources at fine spatial and temporal scales. Avoidance of potential competitors may be possible through small differences in the temporal use of habitats and shared resources [46]. Competitor avoidance is a behavioral strategy that reduces the probability of encounters within the foraging range of potential deadly rivals, thus enhancing the survivorship and fitness of the individual [47, 48]. However, avoidance of competitors is likely to invoke costs, such as a reduction in activity, a reduction in foraging rate or efficiency, or an increase in the use of refugia due to the perceived risk of predation [49–51].

Lions and hyenas appear to reduce some of the competitive effects of a shared diet by hunting prey of different sizes or ages [4, 38, 52, 53]. Lions are able to hunt larger prey than hyenas [54], but large groups of hyenas have adapted to hunting migratory prey populations in the Serengeti with the use of a unique commuting system [55]. Despite the lack of evidence for definite temporal partitioning in the activity periods of lions and hyenas, they exhibit slight differences in periods of activity. In several sites within Southern Africa and Tanzania, hyenas were

active for one continuous period during the night, while lions' were active for two or three periods [14], whereas in the Southern Rift Valley of Kenya, hyenas were active after sunset and from the middle of the night to sunrise, with lions active throughout the night from 22h00 to dawn and after sunrise [56]. Therefore, the two predators may actually be avoiding each other by utilizing the same prey abundant areas but at different times. In addition, food competition among lions and hyenas may be alleviated during periods of high resource availability, such as in the case of ungulate carcasses during anthrax outbreaks [57]. Furthermore, both species employ differences in their hunting behavior, with hyenas mainly hunting large groups of prey and selecting target animals from rushing herds [38], while lions mainly employ stalk-and-ambush tactics of small herds of prey [40]. Thus, lions have the advantage in closed habitats while hyenas are likely to benefit from open habitats due to their cursorial nature. Lions tend to select for habitats with tall grass or steep embankments that promote hunting success and increases the catchability of prey [58–60], while hyenas seemingly appear to be able to utilize any type of habitat as habitat generalists [38, 39].

Although many studies have focused on the factors underlying spatio-temporal patterns in species distributions and resource use, few studies have examined the relationship between predator movement responses to each other to explain spatial overlap patterns. To our knowledge, studies have yet to elucidate fine scale movement and behavioral patterns in sympatric lions and hyenas occurring in large-scale natural systems that shares much of the same resources, as mechanisms facilitating coexistence between these two predator species. This study fills this gap, by analyzing fine-scaled movement data obtained from two different ecosystems subject to seasonal influxes of resources, to discern the behavioral differences that allow lions and hyenas to co-exist. In particular, co-existence was assessed at the arid and mesic extremes of their environment, and the spatiotemporal or behavioral differences in their space use and activity patterns examined.

Data were collected using GPS satellite telemetry collars and activity accelerometers outfitted on lions and hyenas located in both a large fenced National Park and within free-ranging areas of two distinct ecosystems. These data were analyzed with the following objectives in mind: (1) to ascertain the environmental, bioclimatic and social factors influencing the spatio-temporal patterns of lions and hyenas, while accounting for individual variation related to the age, sex, and condition of individuals; and, (2) to determine whether these were primarily influenced by heterospecific or conspecific competitors. We proceeded by first assessing the differences among the range use and proportion of shared space use among lion-hyena dyads between the two species. We then compared and contrasted the movement characteristics and activity patterns of the two species. We subsequently analyzed these movement patterns at various distances to competitors and conspecifics. Finally, we evaluated the relative roles of environmental variables versus inter- and intraspecific interactions in determining lion and hyena spatial distributions and movements. To investigate whether lion and hyena space-use patterns signify avoidance competition, we identified areas across lion and hyena ranges with locations of higher-than-expected revisitation rates or locales of long-duration visits. We then assessed whether these shifted in response to the presence of, or proximity to competitors. We also related these patterns to the distribution of resources at a landscape scale, including anthrax endemic areas in the arid environment.

## Materials and methods

### Ethics statement

Relevant permits required to carry out the research were obtained from the Ministry of Environment and Tourism, Namibia (Research/Collecting Permits 1724/2012, 1834/2013, 1956/

2014) and from the Department of Wildlife and National Parks, Botswana (Research Permit EWT 8/36/4 XXVIII (35)). All animal handling procedures were conducted with the ethical clearance of the Animal Research Ethics Committee of the University of KwaZulu-Natal, South Africa (009/13/Animal), and the Institutional Animal Care and Use Committee of University of California at Berkeley (IACUC Protocol #R217-0512B) and Virginia Tech (IACUC Protocol # 15-012). Namibian specimens were shipped to RSA and Germany under CITES permits for the Regulations of Threatened or Protected Species (Permit/Certificate No. 0045192 and 157940).

## Study area

In brief, the study area covered 19,200km$^2$ within the protected areas of the Southern Africa region: the Etosha National Park, a semi-arid savanna in northern Namibia; the Chobe National Park, Linyanti Conservancy, and the NG32 concession of the Okavango Delta, which comprises the Kalahari floodplains of northern Botswana (Fig 1, see S1 Appendix for additional details on these sites). In the Etosha National Park, certain regions are subject to an influx of seasonal resources from annual anthrax outbreaks [61]. The bacterial pathogen, *Bacillus anthracis*, is endemic as a major disease of various game species [62], and provides a significant subsidy of ungulate carcasses to predators and scavengers [57]. The Chobe-Linyanti region, hereafter "Chobe", experiences a seasonal influx of migratory ungulate prey during the dry season in which they congregate around the perennial river [63]. Situated within the

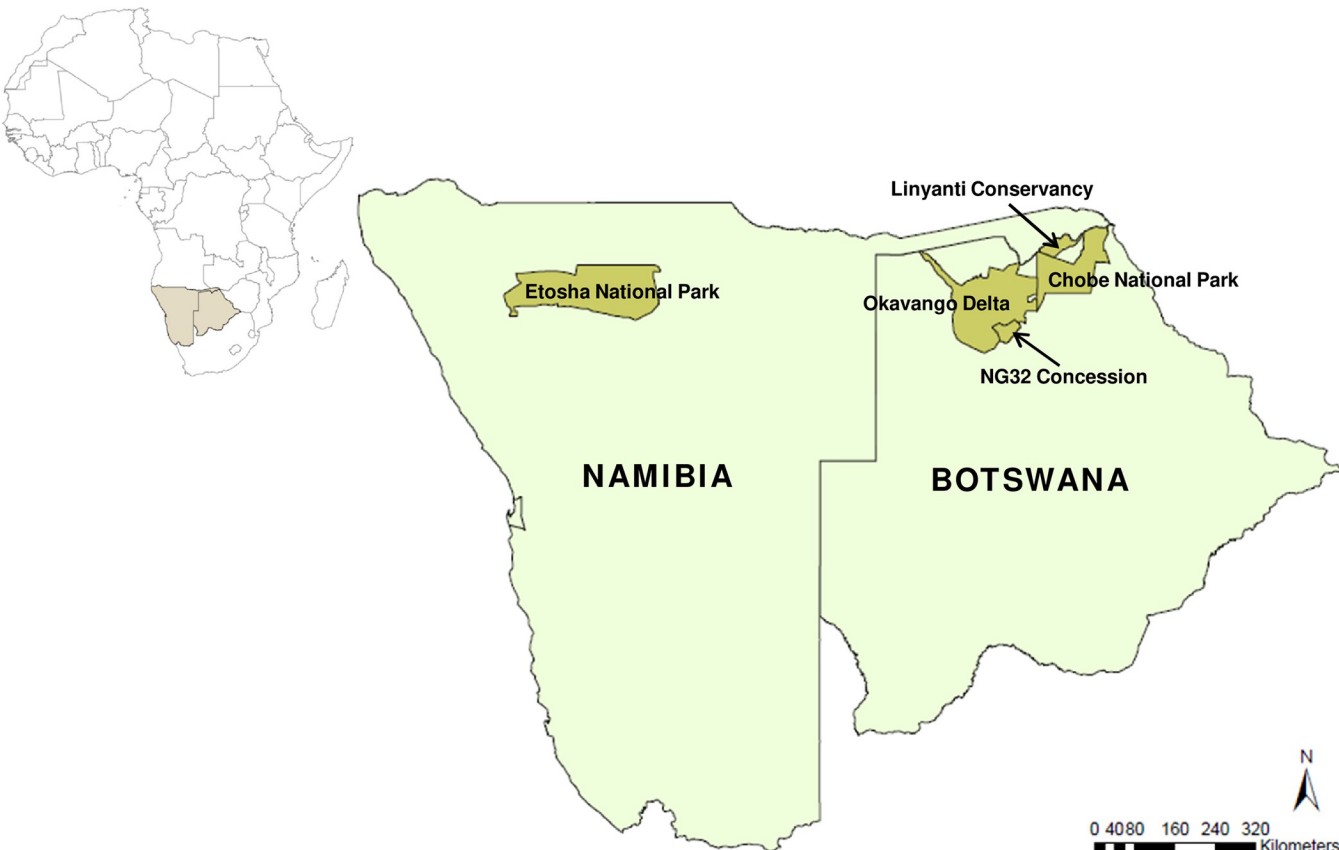

**Fig 1. Location of the study areas.** The map of the African continent shows the countries of Namibia and Botswana shaded, and the protected areas within these countries where the study was conducted. Maps were generated with ArcGIS (ESRI ArcMap v.10.0).

southeastern floodplains of the Okavango Delta, the NG32 concession is subject to an unimodal, annual flood pulse characterized by high variability in interannual flooding [64], and typically comprises a higher prey abundance during the dry season [65]. All study sites, therefore, have a season pulse in prey availability, although through different mechanisms.

## Data collection

A total of 19 lions (13 females and 6 males) and 14 spotted hyenas (10 females and 4 males) were fitted with GPS satellite telemetry collars with dual-axis accelerometers (IridiumTrackM, Lotek Wireless Inc., Newmarket, Ontario, Canada) (see S1 Appendix for additional details of collared animals and capture and sampling protocols). Collars were programmed to record GPS fixes on a schedule that consisted of a fix every 30 minutes during nocturnal periods (18h00 – 6h00 for Etosha individuals, and 17h00 – 8h00 for Botswana individuals), a fix every 5 minutes for two hours twice daily, once after sunset (19h00 – 21h00) and once before sunrise (4h00 – 6h00), and single diurnal fixes both at 10h00 and 14h00. For each datum, activity was averaged from acceleration collected in 8 second bursts over a duration of 240 seconds and given a relative range between 0 and 255 (activity monitor values [AMVs]) to characterize the mean activity/acceleration. Relocation and activity data were downloaded from retrieved collars at the end of the study, with a subset of relocation data obtained from the satellite uplink of unretrieved collars via the Lotek web service (see S1 Appendix for details on data from unretrieved collars).

We retrieved 63% of all possible relocations while collars were deployed, due to the loss of collars from individuals that were killed, or for which we were unable to retrieve the collar ($n$ = 3 lions and 6 hyenas). From the retrieved relocations ($n$ = 575,418) over all study sites, 73% were from lions with 27% from hyenas. This dataset is roughly split between the two ecosystems with 43% of relocations from Etosha and 57% from Botswana. However, our relocation records from Etosha are significantly more complete (99.6% for lions and 84.1% for hyenas) compared to our relocation records from Botswana (53.3% for lions and 35.8% for hyenas). All statistical analyses were conducted in R version 3.5.1 (R Core Team, 2018), and all GIS applications were conducted in ArcGIS (ESRI ArcMap v.10.0, Redlands, CA, USA). See Table 1 for a list of expectations and key results.

## Species range use

The movement dataset of the lions and spotted hyenas were collected over a four-year period split between the two study areas. From this dataset, we removed individuals with less than 30 tracking days, and created different subsets of sampling intervals to ensure for scale appropriateness in subsequent analysis [66]. Sampling intervals occurred every 30 or 5 minutes (depending on the sampling frequency over that sampling period, see above). The regularly sampled data downloaded from the collars all had some incidences of missed fixes (mean ± SE: lion 0.57 ± 0.03%, hyena 0.44 ± 0.06%), while the relocation fixes uploaded via satellites transmitted only a subset of locations during the study period in order to conserve collar batteries. Where collars were unable to be retrieved, movement data was downloaded from the Lotek web service, which consisted of relocations once every third fix of the programmed schedule (15-minute fixes during the 5 minute schedule, and 90 minute fixes during the 30 minute schedule). Thus, the satellite data for some individuals ($n$ = 3 lions, $n$ = 6 hyenas) had missed fixes that ranged between 5 and 1040 minutes. Therefore, for analyses that included satellite individuals, we removed data that had a time difference in GPS fixes greater than 15 minutes from the 5-minute schedule, and greater than 90 minutes from the 30 minute schedule (which tended to occur towards the end of the study as a result of failing batteries). We then

**Table 1. Expected behavioral patterns for lions and spotted hyenas, based on literature, for analyses undertaken in this study.**

| Context | Source | Expected behavioral patterns | | Key results |
|---|---|---|---|---|
| | | Lion | Spotted hyena | |
| Range use | (Boydston et al. 2003a, Kolowski & Holekamp 2009, Périquet et al. 2016, Kittle et al. 2016, Zehnder et al. 2018) | individuals will have larger home ranges and core areas during nocturnal periods and in the wet season, and in arid environments | | • **no differences in sizes of home ranges and core areas across reserves and diel cycles**; • lions had larger home ranges and core areas in wet season; • hyenas had larger core areas in wet season |
| | | home ranges and core areas of lions and hyenas will overlap more in mesic environments | | • **lions and hyenas shared more of their core areas in Etosha than in Chobe**; • lions in Etosha shared more of their home ranges with hyenas than they did with other lions; • **no differences in sizes of home ranges and core areas between species** |
| | | lions will have smaller home ranges and core areas than hyenas | hyenas will have larger home ranges and core areas than lions | |
| Activity patterns | (Kolowski et al. 2007, Hayward & Slotow 2009) | individuals will exhibit increased activity during the night, in the wet season, and with lower activity during the day, in the dry season | | • lions and hyenas had increased activity during the night, **but no differences across seasons**; • hyenas had higher activity than lions |
| Movement characteristics | (Funston et al. 2001, Boydston et al. 2003b, Hopcraft et al. 2005, Stratford & Stratford 2011, Vanak et al. 2013, Périquet et al. 2015a, 2016) | lions will move at slower speeds, have smaller step lengths and net-squared displacements, and have more tortuous movement paths than hyenas | hyenas will move at faster speeds, have larger step lengths and net-squared displacements, and have straighter movement paths than lions | • hyenas moved at faster speeds and had larger step lengths and net-squared displacements than lions; • lions had more tortuous movement paths than hyenas during nocturnal periods; • **Chobe hyenas had more tortuous movements than Chobe lions after sunrise (from 6h00-10h00)** |
| Age | (Funston et al. 2003, Boydston et al. 2005, Elliot et al. 2014) | older individuals will have larger home ranges and core areas | | • **no relationship between age and home ranges or core areas** |
| Gender | (Funston et al. 1998, 2001, Boydston et al. 2001, 2005, Packer et al. 2005) | males to have larger home ranges and core areas than females | females to have larger home ranges and core areas than males | • **no differences between male and female lion home ranges and core areas**; • female hyenas had larger home ranges and core areas than male hyenas during the wet season |
| Body Condition | (Otali & Gilchrist 2004, Kolowski & Holekamp 2008) | individuals with higher conditions will have smaller home ranges / core areas, and move in slower, more tortuous movement paths | individuals with higher conditions will have smaller home ranges / core areas, and move in faster, straighter movement paths | • **no relationship between body condition scores and home range or core areas**; • male lions with low body condition scores moved at higher speeds and had more directional movements than females; **female lions with high body condition scores moved at higher speeds** and had more tortuous movements than males; • hyenas with low body condition scores had more tortuous movements, while hyenas with high body condition scores had more directional movements |
| Group Size | (Cooper 1989, 1991, Holekamp et al. 1997, Mosser & Packer 2009, Loveridge et al. 2009) | individuals from larger groups will have smaller home ranges and core areas | | • **no relationship between pride or clan sizes and home ranges or core areas** |

(*Continued*)

**Table 1.** (Continued)

| Context | Source | Expected behavioral patterns | | Key results |
|---------|--------|------------------------------|---|-------------|
| | | **Lion** | **Spotted hyena** | |
| Anthrax areas | *(Good et al. 2008, Bellan et al. 2012, Dougherty et al. 2020)* | Etosha individuals will shift home ranges to encompass anthrax areas, and will have slower speeds and more tortuous paths in anthrax endemic areas | | • 73% of Etosha lions and 75% of Etosha hyenas shifted their home ranges to the anthrax endemic areas during the wet season; <br> • **no difference in speed or tortuosity of Etosha lions** <br> • Etosha hyenas had more tortuous movements in anthrax areas with higher probabilities of site-attracted foragers, and had more directional movements in areas with low probabilities of site-attracted foragers |
| Lunar cycle | *(Theuerkauf et al. 2003, Packer et al. 2011, Cozzi et al. 2012, Broekhuis et al. 2014)* | individuals will have higher activity, slower speeds, and more tortuous paths in low moonlight conditions | individuals will have higher activity, faster speeds, and straighter movement paths in bright moonlight conditions | • lions had higher activity during low moonlight conditions, and hyenas had higher activity during brighter moonlight conditions; <br> • **Botswana lions had** larger step lengths on new moon nights, with **more tortuous movements on full moon nights**; <br> • Etosha lions and Chobe hyenas had more tortuous movements on new moon nights; <br> **Etosha hyenas** had larger step lengths and **were more tortuous in nights preceding full moon nights** |
| Competitor / conspecific core use areas | *(Cooper 1991, Trinkel & Kastberger 2005, Périquet et al. 2016, Lehmann et al. 2016)* | individuals will have higher activity, faster speeds, straighter paths inside competitor / conspecific core areas, and lower activity, slower speeds, more tortuous paths outside competitor / conspecific core areas | | • Chobe lions and Chobe hyenas had higher activity inside competitor core areas; <br> • hyenas moved at faster speeds inside of competitor core areas; <br> **Etosha lions had more tortuous movements**, while Chobe lions had more directional movements **inside competitor core areas** <br> • Chobe lions were more directional inside conspecific core areas; <br> • **Etosha hyenas had more tortuous movements inside conspecific core areas;** <br> • **Chobe hyenas had lower activity and moved at slower speeds inside conspecific core areas** |
| Distance to nearest competitor / conspecific | *(Cooper 1991, Trinkel & Kastberger 2005, Périquet et al. 2015b, a, Lehmann et al. 2016, Kittle et al. 2016)* | individuals will have higher activity, faster speeds, straighter paths at closer distances to competitors / conspecifics, and lower activity, slower speeds, more tortuous paths at further distances to competitors / conspecifics | | • Chobe hyena had higher activity at closer distances to competitors; <br> • Etosha hyenas moved at faster speeds at closer distances to competitors; <br> • **Etosha lions had more tortuous movements at closer distances to competitors** <br> • **Lions and Etosha hyenas had higher activity, and Etosha lions moved at faster speeds at closer distances to conspecifics than to competitors** |

*(Continued)*

**Table 1.** (Continued)

| Context | Source | Expected behavioral patterns | | Key results |
|---|---|---|---|---|
| | | **Lion** | **Spotted hyena** | |
| Revisitation & Duration (RD) – bioclimatic factors | (Hayward & Hayward 2007, Hayward & Slotow 2009, Cozzi et al. 2012, Schuette et al. 2013, Edwards et al. 2017, Sogbohossou et al. 2018) | individuals will have increased recursivity during the night, in cool temperatures, during the wet season, and will have extended stays during the day, in hot temperatures, and during the dry season | | • individuals had increased recursions during nocturnal periods, and **varied between revisitation and duration across temperatures and seasons** |
| | | individuals will have increased recursivity in low moonlight conditions, and extended stays in bright moonlight conditions | individuals will have increased recursivity in bright moonlight conditions, and extended stays in low moonlight conditions | • **Etosha hyenas had increased recursivity during low and bright moonlight conditions**, with extended duration between new and full moons<br>• **moonlight conditions had no effect on lion RD patterns** |
| Revisitation & Duration (RD) – landscape features | (Kolowski & Holekamp 2009, Valeix et al. 2010, de Boer et al. 2010, Roever et al. 2010, Latham et al. 2011, Bellan et al. 2012, Abrahms et al. 2015, Edwards et al. 2015, Dickie et al. 2016, Davies et al. 2016, Kushata et al. 2018) | individuals will have higher recursions to areas of high NDVI, to anthrax endemic areas, carcass sites, water sources, and roads; and will have extended stays in areas of high NDVI, in anthrax endemic areas, at carcass sites, water sources, and far from roads | | • land cover categories were one of the most important factors explaining for RD patterns in all individuals;<br>• Etosha hyenas had higher recursions to, and longer durations in areas with higher probability of site-attracted foraging ungulates; **Chobe hyenas had higher recursions to locations far from water** and had extended stays in areas closer to water sources |
| Revisitation & Duration (RD) – interactive covariates | (Cooper 1991, Durant 1998, Honer et al. 2002, Funston et al. 2003, Höner et al. 2005, Trinkel & Kastberger 2005, Watts & Holekamp 2008, 2009, Holekamp & Dloniak 2010, Watts et al. 2010, Benhamou et al. 2014, Périquet et al. 2015b, a, 2016, Lehmann et al. 2016, Swanson et al. 2016, Dröge et al. 2017, Kushata et al. 2018) | individuals will have higher recursions to areas far from competitors / conspecifics, in areas with a low probability of encountering competitors / conspecifics, and inside competitor / conspecific core areas; and will have extended stays in areas far from competitors / conspecifics, in areas with a low probability of encountering competitors / conspecifics, and outside competitor / conspecific core areas | | • **female lions and hyenas** had longer durations in areas of low competitor probabilities, at far distances to competitors and outside competitor core areas, and **had increased recursions to areas of high competitor probabilities, at close distances to competitors and inside competitor core areas;**<br>• **male lions had longer durations in areas of high competitor probabilities**<br>• **lions were equally split between revisitation and duration in areas of high and low conspecific probabilities;**<br>• **hyenas had longer duration in areas of high conspecific probabilities**, and had higher recursions in areas of low conspecific probabilities, at far distances to conspecifics, and outside conspecific core areas |

applied the "ctmm" R package [67] to fill in the missing coordinates in the schedule where required.

The data for analyses comprised of relocations from both collared individuals and relocations uploaded via satellite filled in with the ctmm method. From these data, nocturnal periods of relocations every 30 minutes were organized to include only fixes obtained during times of 18h00–6h00 or 17h00–8h00, as well as split into dry and wet seasons for the construction of home ranges and core areas. We filtered out 4-hour locations over a 24-hour period

(maximum 6 fixes per day) from the relocation data, which were also split into dry and wet seasons to use for a comparison of nocturnal and diurnal ranges.

We constructed two types of utilization distributions (UDs) for each individual's overall and seasonal ranges. First, we used the adehabitatHR kernel density estimator (KDE) with the reference bandwidth as the smoothing factor [68]. Second, we used the *a*-LoCoH (local convex hulls) adaptive method [69, 70], which facilitates the identification of regularly revisited sites, such as dens, waterholes, or riverbanks [69]. We used the 95% and 50% UDs (both methods) to represent respectively the home ranges and core use areas of individuals. We used the *t*-test in R v.3.5.1 to compare the sizes of home ranges and core areas among competitors (lions and hyenas) with regards to location (Etosha versus Chobe), seasons, and segments of diel cycles. Within the kernel density ranges and utilization distributions, we used the intersection function from the "rgeos" R package [71] to compute the areas of overlap between neighboring individual ranges, and used the *t*-test to determine the significance of differences in species' overlapping ranges among ecosystems, across seasons and diel cycles.

### Species movement and activity patterns

To analyze the movement patterns of lions and hyenas, we used the "adehabitatLT" R package [68] to calculate the step lengths and turning angles between successive locations. We calculated the speed and distance travelled (step length), path tortuosity (turning angle), and net-squared displacement (NSD) for each individual and combined these individuals into species groups. Since turning angles range continuously on a circular scale from -180 to 180, we used the "CircStats" R package [72] to calculate the vectorized mean turning angle, which treats each observation as a vector on the unit circle to indicate the direction of the resultant vector [73]. Note that for CircStats, Watson's two sample test is significant at $p < 0.10$. We then compared the seasonal, sex-specific movement metrics of lions and hyenas from each of the two ecosystems. We also assessed these differences over various times of the day, between diel cycles, land cover types, and across seasons.

We further analyzed the seasonal activity patterns of lions and hyenas according to the full moon and new moon phases of the lunar cycle, as in Cozzi et al. [27]. We defined full moon nights when ≥95% of the lunar disc was illuminated and new moon nights were when moonlight intensity was ≤5%. For each day during full moon and new moon phases, we divided each 24-hour period into seven different sections (afternoon, dusk, night, nadir, night-end, dawn, morning) to reflect the main activity periods for lions and hyenas [14]. We used "suncalc" in R [74] to calculate the times of periods for each day with dusk lasting from sundown to the end of evening astronomical twilight and dawn from the end of morning astronomical twilight to sunrise. The period between the end of evening twilight to the beginning of morning twilight was divided into three equal intervals in minutes to reflect night, nadir, and night-end. The day period from sunrise to sundown was divided from noon into morning and afternoon periods. We then calculated the proportion of the averaged activity measures that occurred during each of these periods.

For those individuals that had relocations within the core use area (or 50% isopleth) of either competitors or conspecifics, we assessed whether such proximity had any effect on the activity patterns as well as the speed and tortuosity of the focal animal. We assigned a value of 1 to each individual's location that occurred within the core use area of a competitor's or conspecific's UD and assigned a value of 0 to each individual's location not within a competitor or conspecific core use area. We averaged the activity measures and movement metrics for each individual occurring inside and outside the core use areas of competitors and conspecifics. We then tested whether the average activity and movement metrics of lions and hyenas differed

significantly from when they were inside or outside of the core use areas of competitors or conspecifics.

## Inter- and intra-specific effects

**Frequency of time-matched distances.** The distances between collared individuals were obtained to analyze for interactive effects. We developed a temporally aligned matrix for each individual that overlapped in their collaring periods. For each sampling record of collar overlap, we measured the minimum Euclidean distance of that individual to all other collared individuals, at all locations, over the same time period. For each individual, we calculated the percentage frequency occurrences of time-matched distances between the individual to any heterospecific or conspecific competitor, in five frequency bins between distances of 5 km, 1 km, 200 m, 100 m, 50 m, and 10 m. We determined whether lions or hyenas occurred at closer distances more often to each other (heterospecific competitors) than with one another among their own species (conspecific competitors).

**Consecutive time points.** In addition, we ascertained the length of time individuals spent with either competitors or conspecifics for all dyads that occurred at distances $\leq$ 5 km. We used the *distm* function from the "geosphere" R package to calculate the distance for each time-matched point in the trajectory of successive fixes for each dyad. We then calculated the number of consecutive time points (indicating a longer duration together) for which individuals of the dyad were at a distance below a given value (2 km, 1 km, 500 m, 200 m, 100 m). We again determined whether either lions or hyenas occurred at closer distances more often to heterospecific or conspecific competitors, during consecutive time points.

**Interactive variables.** We included interactive variables to examine the influence of competitors and conspecifics on the rates of revisitations and visit durations of lions and hyenas within their ranges. We measured distances between collared individuals and constructed GIS layers representing areas with a probability of competitor and conspecific use to analyze for interactive effects. To obtain the distances between collared individuals, we developed a temporally aligned matrix of each individual that overlapped with each other during the collaring period. For each sampling record of collar overlap, we measured the minimum Euclidean distance of that individual to all other collared individuals at all locations over the same times.

Individual seasonal UDs were overlaid, and pixel cell values averaged to generate a total combined lion UD and a total combined hyena UD for each of the dry and wet seasons. We converted the combined kernel UDs to volume UDs to obtain probability of use values for each cell. We subtracted volume UD values from 100 as in Kittle et al. [75], to obtain a more intuitive value with low use cells reflected by low values and high use cells reflected by high values. UD pixel values were then extracted and assigned to each locational point as a probability of competitor use area in ArcGIS v.10.0. We also repeated the process as above for a probability of conspecific use area. To construct the layer of the potential conspecific range, we overlaid the seasonal UDs of all other individuals of the same species, while excluding the individual the layer was being created for.

The Etosha lion UD from the period of 2013 to 2015 was constructed from 11 individuals of 10 prides with 52,547 relocations for the dry season and 65,428 relocations for the wet season. The Chobe lion UD from the period of 2015 to 2017 was constructed from 6 individuals of 5 prides with 30,016 relocations for the dry season and 43,356 relocations for the wet season. Etosha spotted hyena dry season UD was constructed from 57,211 relocations of 8 individuals from 8 clans, with wet season UD from 70,724 relocations of 7 individuals from 7 clans during the years of 2013 to 2015. Chobe spotted hyena dry season UD was constructed from 14,546

**Table 2. Description of ecogeographical variables (EGVs) used for cluster analyses.**

| Ecogeographical Variables | Description | Data Resolution/Type |
|---|---|---|
| *Bioclimatic variables* | Time of day | each relocation |
| | Temperature | each relocation |
| | Season | binary variable |
| | Moon illumination | probability index, 0-1 |
| | Precipitation amounts | pentad average |
| *Landscape features* | Slope | 30m x 30m |
| | Land cover categories | 30m x 30m |
| | NDVI | 30m x 30m, seasonal mean |
| | Anthrax carcass probability | UD value |
| | Distance to nearest available carcass | Euclidean, in m |
| | Distance to nearest permanent water source | Euclidean, in m |
| | Distance to nearest seasonal water source | Euclidean, in m |
| | Distance to nearest road | Euclidean, in m |
| *Human disturbances* | Distance to nearest anthropogenic feature | Euclidean, in m |
| *Interspecific interactions*[a] | Distance to nearest competitor | Euclidean, in m |
| | Competitor probability | UD value |
| | Competitor core | 1 = inside core, 0 = outside core |
| *Intraspecific interactions*[a] | Distance to nearest conspecific | Euclidean, in m |
| | Conspecific probability | UD value |
| | Conspecific core | 1 = inside core, 0 = outside core |

EGVs were included in cluster analyses of the revisitation and duration (RD) space for each lion and spotted hyena individual within Etosha National Park, Namibia and the Chobe National Park, Linyanti Conservancy, and Okavango Delta[b], Botswana.

[a]*Not included in cluster analyses of lions from Okavango Delta, Botswana.*

[b]*No spotted hyenas were collared from the Okavango Delta, Botswana.*

relocations of 4 individuals from 4 clans, with wet season UD from 15,335 relocations of 5 individuals from 5 clans during the years of 2015 to 2017.

**Ecogeographical variables.** Ecogeographical variables (EGVs) such as distance to water, land cover types, and precipitation that have been statistically associated with species distribution [76–78], and were attached to each point within the RD space to evaluate their influences on lion and hyena revisitations and visit durations (Table 2). We used ArcGIS (ESRI ArcMap v.10.0, Redlands, CA, USA) to measure the minimum Euclidean distance from all location points to various geographical features and landscape attributes, including distance to carcasses in Etosha (see S1 Appendix for additional details on the collection of Etosha carcass data).

Digital elevation maps of the four study areas were obtained from Landsat 8 images, courtesy of the U.S. Geological Survey, using a spatial resolution of 30 m. We derived the slope and aspect from these digital elevation maps in ArcGIS. Open-source land cover maps generated from LandSat thematic mapper data were downloaded for Namibia and Botswana (2010 Scheme II) via the RCMRD GeoPortal (http://geoportal.rcmrd.org), to which we assigned arbitrary values to reflect discrete land cover categories. We used Google Earth Engine's [79] Normalized Difference Vegetation Index (NDVI) 8 day composites from the duration of the study period to obtain the mean NDVI values for each of the four study areas for each of the dry and wet season. As a measure of vegetation productivity [80–83], NDVI has been used as an index of prey availability [78, 84], with areas of increased vegetative cover correlated to increased ungulate and herbivore biomass [85–89]. These Landsat 8 composites are generated from Tier

1 orthorectified scenes, using the computed top-of-atmosphere reflectance. All the images from each 8-day period are included in the composite, with the most recent pixel as the composite value. NDVI values ranged from -1 to 1, with negative values corresponding to clouds and water, values near zero representing rock and bare soil, moderate values of 0.2 – 0.3 representing shrub and grassland, with high values close to 1 indicating temperate forests and tropical rainforests. Using ArcGIS, we calculated the mean center for each individual's range, from which we obtained the relevant sunrise/set and moonrise/set times (Astronomical Applications Department of the U.S. Naval Observatory, from https://aa.usno.navy.mil), and interpolated the average precipitation from available CMAP Precipitation data (Climate Data and Resources, NOAA/OAR/ESRL PSD, Boulder, Colorado, USA, from https://www.esrl.noaa.gov.psd/). The "lunar" package in R version 3.5.1 (R Core Team, 2018) was used to assign moon phases and moon illumination values (ranging from 0 = new moon to 1 = full moon) to all locations, according to each location's unique timestamp. For any location points collected between moonset and moonrise, the moon illumination was assigned a false logical vector.

## Time-use metrics

To assess how lions and hyenas adjust their movements over time and across seasons, we quantified the time-use metrics (i.e., revisitation rates and visit duration) from the T-LoCoH hull parent points for each individual's seasonal trajectories. We used the "T-locoh.dev" R package [70] to identify sites of repeated visits (nsv, number of separate visits to each cell) and sites of average visit durations (mnlv, mean number of locations per visit to each cell). A lion and hyena whose ranges overlapped in Etosha is shown in Fig 2A, with all other individuals shown in S16 and S17 Figs. As we were interested in the period of time which both lions and hyenas overlap in their activity periods [14], we chose to use an inter-visit gap (IVG) of 12 hours (one nocturnal period) to distinguish locations with more than 12 hours of time between them as separate visits. From this, we created density plots of each individual's time use metrics on the landscape, which we refer to as revisitation and duration (RD) space (Fig 2B). We then obtained the local values of various ecogeographical and interactive variables (see below) and associated these covariate vectors to each point within the RD space.

We used a factor analysis of mixed data (FAMD) to test for a statistically significant relationship between our selected ecogeographical and interactive variables with points in our constructed RD space. From the principal dimensions identified by our FAMD that describe >80% of the cumulative variation, we chose the variables with the highest scores in each dimension as the most important covariate. From this analysis, we determined the appropriate covariate combinations prior to clustering. We then performed a cluster analysis on the points within the RD space for each individual. We built mixed data type cluster models according to the FAMD results, and applied three different clustering algorithms for 2-8 clusters. We used a *k*-prototypes clustering algorithm [90] from the "clustMixType" R package, which is based on *k*-means for mixed type data. Due to the stochasticity of the *k*-prototype algorithm, each model configuration was recomputed 50 times with random initializations to obtain a model of minimum total distance. We also calculated a dissimilarity matrix using the "gower" metric in the daisy function from the "*cluster*" R package, which we used in the agglomerative hierarchical and PAM (partitioning around medoids) [91] clustering algorithms for further clustering. We color-coded the points on the map according to the clustered results, and then examined whether the points within a range of revisitation (R) and duration (D) values from the RD space fell mainly within the identified clusters.

We visually inspected the results of the clustering analysis and determined the appropriate clustering method. We chose the clustering method according to the distribution of the

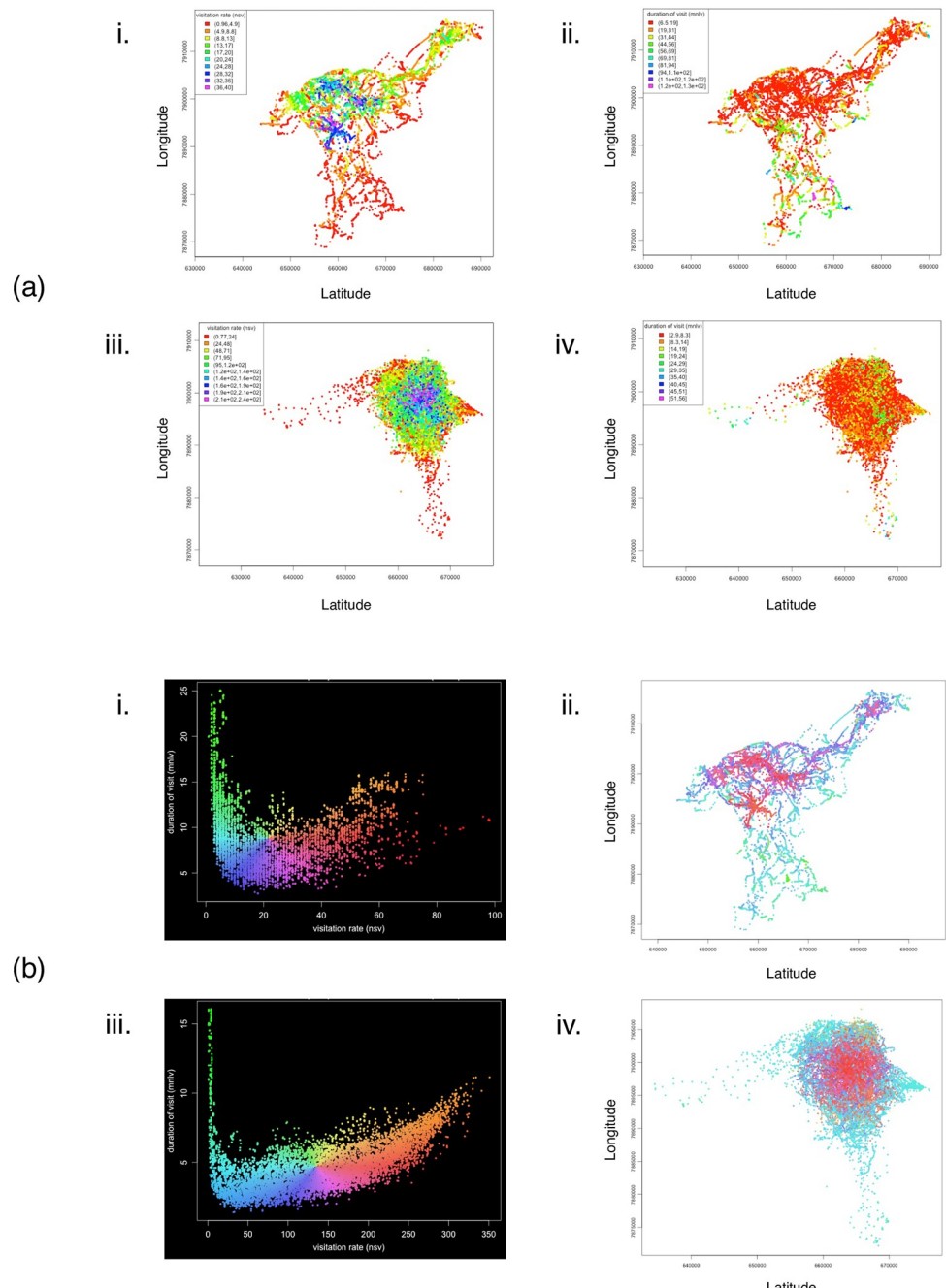

**Fig 2. Revisitation and duration (RD) space plots indicated from the hull parent points of a lion and spotted hyena whose ranges overlapped.** (a) Hull parent points for a collared female lion (NU-33865, i & ii, both figures) and female spotted hyena (GO-33869, iii & iv, both figures) whose ranges overlapped in Etosha. Parent points are colored by visitation rate (nsv, number of separate visits; i & iii), and duration of visit (mnlv, mean number of locations in the hull per visit; ii & iv). (b) RD space scatterplots (i & iii) with X-axis = visitation rate (nsv), and Y-axis = duration of visit (mnlv), provide a legend for revisitation/duration (RD) values for the maps (ii & iv). Points in the RD space have been jiggled to better see point density, and each point within the RD space represents a hull (i) lion $n = 9160$, (iii) hyena $n = 11898$. Points on the maps (ii & iv) are colored by their location in the RD space. Separate visits defined by an inter-visit gap period $\geq$ 12 hours. Hulls were created using the adaptive method. Duplicate points are offset by 1 map unit.

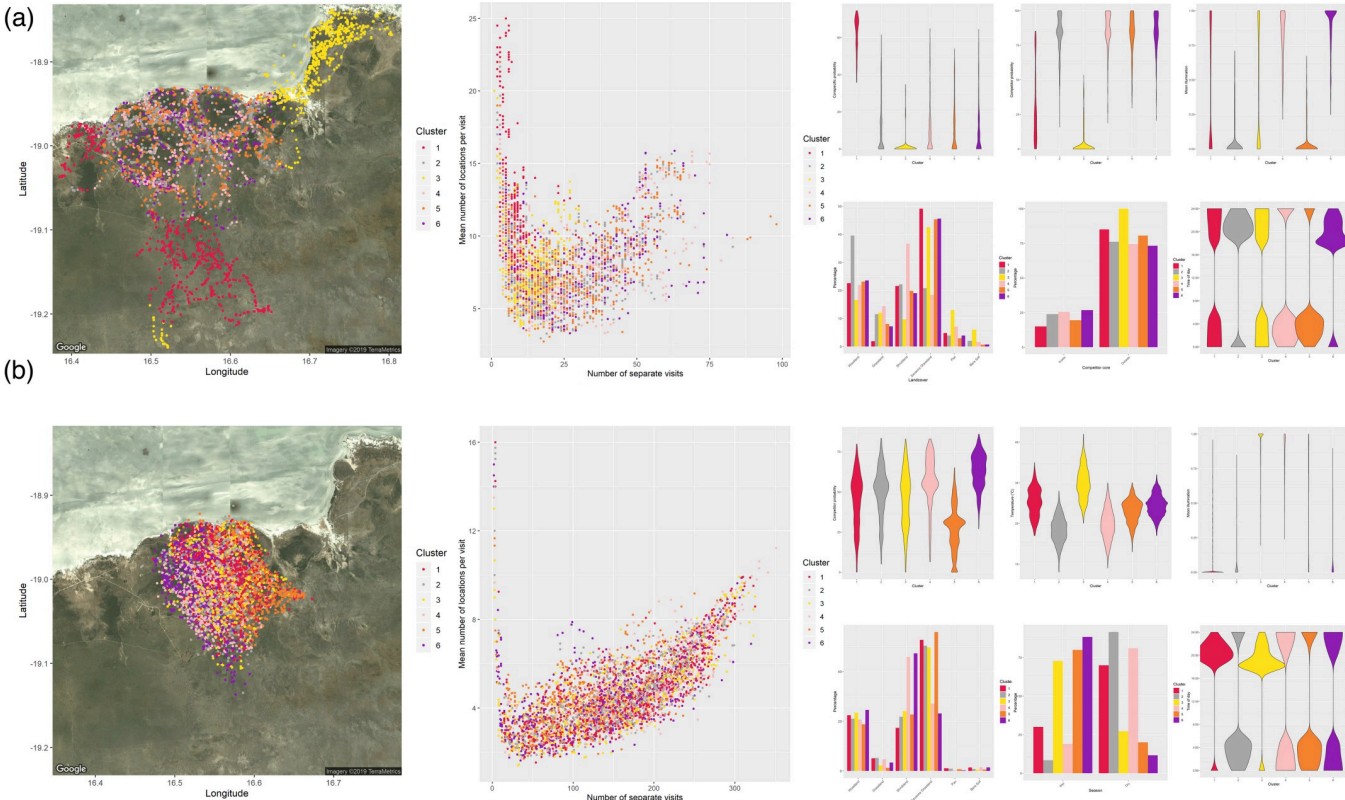

**Fig 3. Cluster analyses of the revisitation and duration of a lion and spotted hyena whose ranges overlapped.** Maps (far left panels) depict the individual relocations of a collared (a) lion (NU-33865) and (b) spotted hyena (GO-33869) whose ranges overlapped in Etosha. Relocations are color-coded according to the clusters indicated by the range of revisitation (number of separate visits) and duration (mean number of locations per visit) values in RD space plots (central panels). Clusters in the RD space were determined with the *k*-prototype algorithm and are based on ecogeographical variables attached to each relocation. The smaller plots (right panels) present the distribution and percent category of clusters for each of the ecogeographical variables selected from the factor analysis of mixed data (FAMD).

FAMD selected covariates for each cluster and the percentage of categories occurring in each cluster. We chose the *k*-prototype clustering algorithm because it resulted in more defined clustered groups within the individual's RD space, and it was more distinctive in the distribution of the clusters according to the covariates. An example is shown in Fig 3 for a lion and hyena whose ranges overlapped in Etosha, with all other individuals shown in S18 and S19 Figs. We used the multivariate *t*-distribution algorithm from the "ggplot" R package to draw ellipses around the clustered points, and subsequently chose the number of clusters according to how the points were clustered together in the RD space.

# Results

## Species range use

Lions and hyenas used the same types of habitat and occurred most frequently within similar land cover types between the two ecosystems. In Etosha, both species demonstrated higher frequencies of relocations within grassland habitats, whereas in Botswana sites, the most prevalent land cover types utilized by both species were shrublands (S1 Table). Kernel density UDs of lion and hyena home ranges exhibited a high degree of spatial overlap within both Etosha and Chobe (Fig 4). Seasonal UDs for all individuals are presented in S2 and S3 Figs and S2 Table.

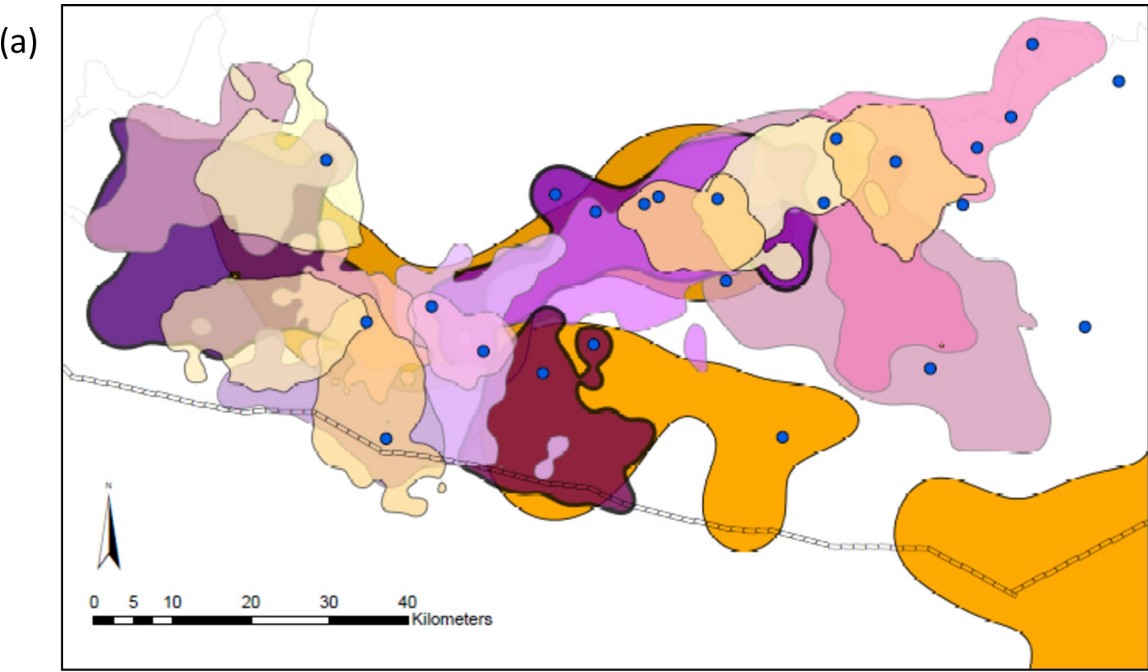

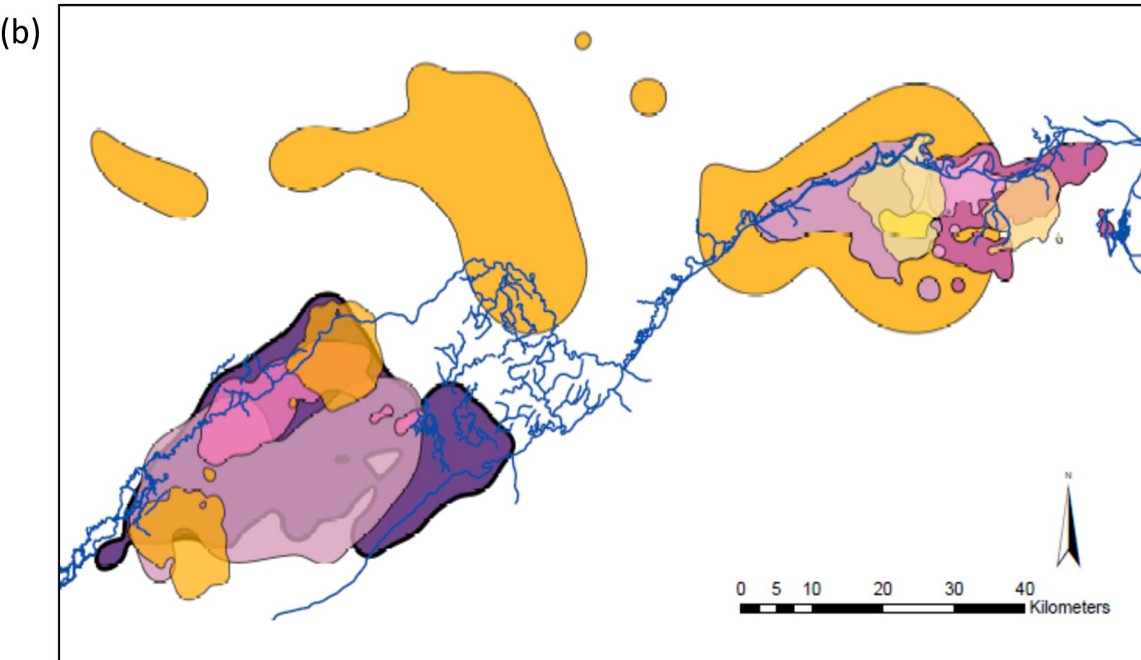

**Fig 4. Overlapping ranges within study areas as represented by 95% kernel contours.** Polygons of dark purple shades with black outlines = male lion ranges. Polygons of lighter purple/pink shades with grey outlines = female lion ranges. Polygons of orange/yellow shades = spotted hyena ranges. Overlapping polygons are set to 20% opacity for easier visualization. (a) Etosha site with the pan to the north and the park's fence along the southern boundary. Blue dots represent permanent water points. (b) Chobe/Linyanti site with the river separating Botswana to the south and Namibia to the north. The river is indicated by blue lines.

Lion and hyena home ranges constructed with the kernel density estimates were larger than the *a*-LoCoH estimates, although not significantly so (Table 3). Using the time-scaled distance measure incorporated into the UDs, there were no significant differences in the sizes of either

**Table 3. Lion and spotted hyena range sizes (km$^2$).**

**(a)**

| Species | Region | | Home range sizes (km$^2$) | |
|---|---|---|---|---|
| | | | KDE | a-LoCoH |
| Lion | ENP | | 577.17 ± 93.90 | 361.96 ± 48.36 |
| | CNP | | 363.33 ± 119.81 | 257.31 ± 101.13 |
| Spotted hyena | ENP | | 718.56 ± 327.89 | 413.53 ± 115.07 |
| | CNP | | 478.68 ± 354.92 | 194.08 ± 82.73 |

**(b)**

| Range | Period | | Species range sizes (km$^2$) | |
|---|---|---|---|---|
| | | | Lion | Spotted hyena |
| Home range | Nocturnal | | 327.82 ± 49.61 | 331.49 ± 90.25 |
| | Diurnal | | 287.84 ± 49.81 | 257.47 ± 74.88 |
| Core use area | Nocturnal | | 98.98 ± 17.11 | 74.61 ± 15.84 |
| | Diurnal | | 95.64 ± 19.65 | 54.47 ± 12.08 |

| | | Seasonal range sizes (km$^2$) | | | |
|---|---|---|---|---|---|
| | | Dry | Wet | Dry | Wet |
| Home range | | 157.76 ± 36.36 | 253.56 ± 45.10 | 131.13 ± 25.18 | 348.25 ± 97.40 |
| | Nocturnal | **162.45 ± 36.86** | **278.39 ± 41.36** | 154.42 ± 22.99 | 383.96 ± 123.69 |
| | Diurnal | **121.15 ± 29.72** | **244.36 ± 42.33** | 108.12 ± 23.33 | 293.02 ± 95.53 |
| Core use area | | 41.99 ± 10.86 | 70.04 ± 11.39 | **30.23 ± 6.93** | **90.54 ± 24.35** |
| | Nocturnal | **45.32 ± 11.65** | **106.14 ± 21.40** | 48.16 ± 8.75 | 110.27 ± 36.21 |
| | Diurnal | **42.23 ± 10.71** | **95.69 ± 23.00** | 30.73 ± 8.65 | 73.51 ± 21.98 |

| | | Protected area range sizes (km$^2$) | | | |
|---|---|---|---|---|---|
| | | ENP | CNP | ENP | CNP |
| Home range | | 361.96 ± 48.36 | 257.31 ± 101.13 | 413.53 ± 115.07 | 194.08 ± 82.73 |
| | Nocturnal | 393.45 ± 63.09 | 245.78 ± 72.95 | 426.53 ± 124.31 | 179.44 ± 104.26 |
| | Diurnal | 336.57 ± 62.91 | 226.94 ± 79.00 | 330.93 ± 102.30 | 139.94 ± 93.95 |
| Core use area | | 113.31 ± 15.93 | 56.96 ± 21.02 | **85.21 ± 13.99** | **38.14 ± 21.56** |
| | Nocturnal | **127.80 ± 24.42** | **62.95 ± 17.53** | 92.65 ± 22.65 | 45.74 ± 13.53 |
| | Diurnal | 104.74 ± 18.03 | 84.28 ± 39.43 | 70.30 ± 15.45 | 29.14 ± 14.50 |

(a) Home range sizes (mean ± SE) were estimated with the kernel density estimate and a-LoCoH method. (b) Diel and seasonal home ranges and core use areas (mean ± SE) were estimated with the a-LoCoH method. Lion and spotted hyena ranges were from the Etosha National Park, Namibia (ENP) and the Chobe National Park, Linyanti Conservancy and Okavango Delta[a], Botswana (CNP). Values in bold indicate a significant difference between seasons or between protected areas for each species, and greyed out values were not significantly different.

[a]*No spotted hyenas were collared from the Okavango Delta, Botswana.*

a-LoCoH estimated 95% home ranges and 50% core use areas between lions and hyenas during nocturnal and diurnal periods (Table 3, all t-tests, $p > 0.05$). Overall, lions had larger home ranges and core areas in the wet season than they did in the dry season for each of the nocturnal (home range: $t = -2.09$, df = 30.8, $p < 0.05$; core area: $t = -2.50$, df = 24.7, $p < 0.05$) and diurnal (home range: $t = -2.38$, df = 28.7, $p \ll 0.05$; core area: $t = -2.11$, df = 22.6, $p < 0.05$) periods (Table 3), whereas hyenas only had larger core areas in the wet season (Table 3, $t = -2.38$, df = 12.8, $p < 0.05$). Additionally, there were no significant differences in the sizes of either home ranges or core areas between lions and hyenas in each reserve, regardless of seasons and circadian cycles (Table 3, all t-tests, $p > 0.05$). However, when comparing

**Table 4. Percentage range overlaps in the home ranges and core use areas between lions and spotted hyenas, lion conspecifics, hyenas and lions, or hyena conspecifics.**

| | Home range | | | | Core use area | | | |
|---|---|---|---|---|---|---|---|---|
| | ENP | | CNP | | ENP | | CNP | |
| Lion-Hyena | 20 ± 4% * | | 23 ± 5% | | **8 ± 2%** | | **3 ± 1%** | |
| Lion-Lion | **10 ± 2%** * | | **26 ± 7%** | | 6 2% | | 13 5% | |
| Hyena-Lion | 17 ± 3% | | 23 ± 4% | | 8 ± 2% | | 1 ± 0.5% | |
| Hyena-Hyena | 13 ± 4% | | 36 ± 16% | | 5 ± 3% | | 30 ± 15% | |
| | SEASON | | | | | | | |
| | Dry | Wet | Dry | Wet | Dry | Wet | Dry | Wet |
| Lion-Hyena | 16 ± 5% | 25 ± 5% | 25 ± 8% | 22 ± 6% | 3 ± 2% | *14 ± 4%* | 4 ± 2% | 2 ± 1% |
| Lion-Lion | 9 ± 3% | 11 ± 4% | 27 ± 11% | 25 ± 9% | 5 ± 2% | 7 ± 4% | 11 ± 7% | 16 ± 8% |
| Hyena-Lion | 14 ± 4% | 21 ± 5% | 21 ± 6% | 26 ± 7% | 6 ± 3% | 11 ± 4% | 1 ± 1% | 0.1 ± 0.1% |
| Hyena-Hyena | 7 ± 5% | 21 ± 6% | 42 ± 25% | 30 ± 23% | 1 ± 1% | 8 ± 6% | 34 ± 21% | 27 ± 24% |

Percentage (mean ± SE) range overlaps were from the Etosha National Park, Namibia (ENP) and the Chobe National Park and Linyanti Conservancy, Botswana (CNP). Values in bold indicate a significant difference between protected areas; italicized values indicate a significant seasonal difference; and values with an asterisk indicate a significant difference between heterospecific and conspecific competitors. Greyed out values were not significantly different.

conspecifics across reserves, Etosha lions had larger nocturnal core areas than Botswana lions (Table 3, $t$ = -2.16, df = 15.4, $p < 0.05$). Similarly, Etosha hyenas had larger core areas than their counterparts in Chobe (Table 3, $t$ = -2.77, df = 10.9, $p < 0.05$).

The total area of overlap in the home ranges and core areas of the two predators, both the other's main competitor, are also presented for each pair in S3 Table. We calculated the proportion of lion home ranges and their core areas overlapped by hyena home ranges and core areas, as well as the proportion of hyena home ranges and their core areas overlapped by lion ranges and core areas (S4 and S5 Tables). Lions and hyenas from both Etosha and Chobe shared parts of their home ranges and core use areas (Table 4). However, lions shared more of their core use areas with hyenas in Etosha than they did in Chobe (Table 4, $t$ = 2.19, df = 81.0, $p < 0.05$). In addition, lions in Etosha shared more of their core use areas with hyenas during the wet season than the dry season (Table 4, $t$ = -2.31, df = 34.7, $p < 0.05$). Although lions did not differ in the sizes of overlapped areas between conspecifics or competitors, Etosha lions shared more of their home ranges with competitors than with conspecifics (Table 4, $t$ = -2.35, df = 103.7, $p < 0.05$). By contrast, lion conspecifics in Chobe shared significantly more of their home ranges than they did in Etosha (Table 4, $t$ = -2.19, df = 28.9, $p < 0.05$). Furthermore, individual core areas were more distant from competitor core areas than the nearest territory boundaries of competitors'. This was significant for most cases (all $t$-tests, $p < 0.05$, S6 Table), except for hyenas to heterospecific competitors in Etosha, and to conspecific competitors in Botswana (S6 Table).

## Species movement and activity patterns

Descriptive analyses of lion and spotted hyena movement descriptors are presented in S2 Appendix and S7–S11 Tables. Despite being temporally aligned with the activity periods at night, hyenas exhibited nearly twice the activity rates of lions ($t$ = -2.90, df = 7.2, $p < 0.05$; Fig 5; Table 5; S7 Table). They also moved at characteristically higher speeds than lions in both ecosystems and had greater nocturnal mean step lengths (Table 5; all $t$-tests, $p < 0.001$; S2 Appendix). Hyenas were consistent in that they moved at significantly faster speeds and travelled significantly further than lions throughout different time periods and across both

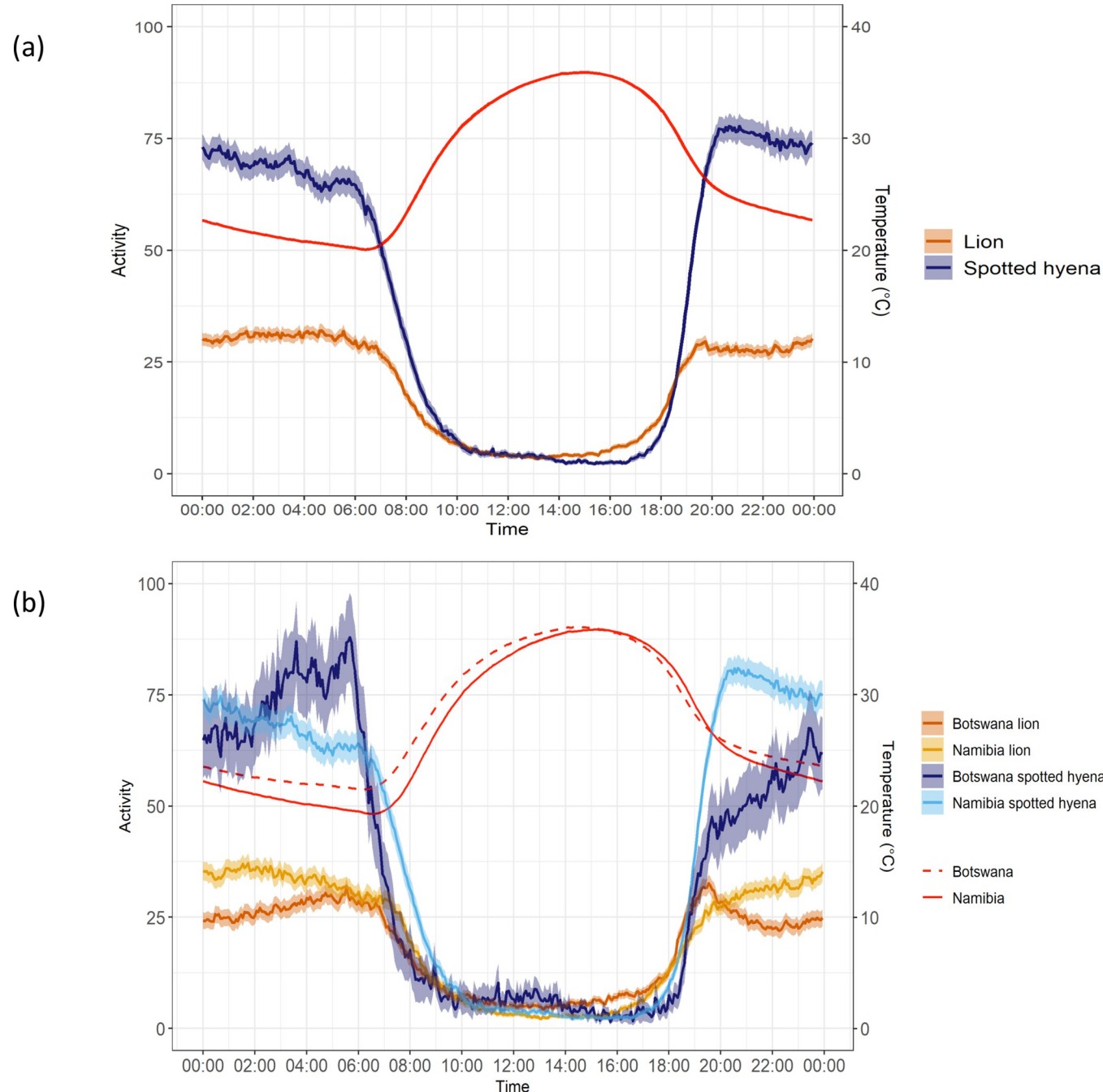

**Fig 5. Mean activity of lions and spotted hyenas.** Activity rates over the 24-hour cycle of (a) lions and spotted hyenas, and (b) lions and spotted hyenas from the Etosha National Park, Namibia and the Chobe National Park, Linyanti Conservancy, and the Okavango Delta[a], Botswana. Means are represented by solid lines with 95% confidence intervals the shaded bars. The red lines indicate the average temperatures (°C) over the 24-hour cycle. [a]*No spotted hyenas were collared from the Okavango Delta, Botswana.*

ecosystems (Table 5; all *t*-tests, $p < 0.05$; S2 Appendix; S7 Table), except for during the wet season in Chobe.

The activity levels of lions and hyenas, as measured by the accelerometers, were significantly affected by temperature, for each hour of the day (S20 Fig). Specifically, lion activity decreased with increasing temperatures for 20 hours of the 24-hour period (adjusted $R^2$ ranging from

**Table 5. Lion and spotted hyena movement metrics from protected areas.**

**(a)**

| Metric | Lion | Spotted hyena |
|---|---|---|
| Activity (AMVs in 24-hour period) | 21.88 ± 41.2 | 40.20 ± 67.8 |
| Nocturnal step length from 30min fixes (m) | 288.90 ± 423.59 | 618.12 ± 684.50 |

**(b)**

| Metric | | ENP | | CNP | |
|---|---|---|---|---|---|
| | | Lion | Spotted Hyena | Lion | Spotted Hyena |
| Speed (m/s) | | 0.187 ± 0.25 | 0.364 ± 0.39 | 0.128 ± 0.23 | 0.310 ± 0.35 |
| 24-hour step length from 4hr fixes (m) | | 1257 ± 1535 | 2009 ± 2156 | 898 ± 1159 | 1744 ± 1857 |
| Nocturnal step length from 30min fixes (m) | | 308 ± 462 | 649 ± 685 | 208 ± 340 | 529 ± 570 |
| Dusk/dawn step length from 5min fixes (m) | | 55 ± 99 | 130 ± 148 | 44 ± 78 | 119 ± 153 |
| 24-hour path tortuosity (radian) (4hr fixes) | Combined seasons | 0.080 ± 2.07 | -2.852 ± 2.39 | 0.432 ± 2.41 | 2.963 ± 2.18 |
| | Dry season | 0.070 ± 2.07 | -2.012 ± 2.24 | 0.296 ± 2.33 | 2.660 ± 2.03 |
| | Wet season | -0.022 ± 2.07 | 1.637 ± 2.60 | 0.209 ± 2.35 | -2.942 ± 2.01 |
| Nocturnal path tortuosity (radian) (30min fixes) | Dry season | 0.018 ± 1.89 | 0.027 ± 1.84 | 0.068 ± 2.44 | -0.018 ± 1.87 |
| Dusk/dawn path tortuosity (radian) (5min fixes) | Combined seasons | 0.053 ± 2.43 | 0.013 ± 1.59 | 0.847 ± 1.92 | 0.001 ± 1.31 |
| | Wet season | 0.097 ± 2.42 | 0.012 ± 1.54 | 1.062 ± 1.85 | 0.004 ± 1.25 |

Additional metrics are available in S7 Table. (a) all lions and spotted hyenas, and (b) from protected areas (PAs) including the Etosha National Park, Namibia (ENP) and the Chobe National Park, Linyanti Conservancy, and the NG32 concession of the Okavango Delta[a], Botswana (CNP). Movement metrics consist of the means ± standard deviations for activity (activity monitor values [AMVs]), step length (m), speed (m/s), and path tortuosity (radian). Values in bold indicate a significant difference between heterospecific competitors within PAs, and italicized values indicate a significant difference between conspecifics across PAs. Greyed out values were not significant.

[a]*No spotted hyenas were collared from the Okavango Delta, Botswana.*

0.012 to 0.220, $p < 0.05$). Similarly, hyena activity decreased with increasing temperatures for 22 hours of the 24-hour period (adjusted $R^2$ ranging from 0.014 to 0.502, $p < 0.05$; S20 Fig).

During 24-hour periods, hyenas demonstrated more directional movements in the semi-arid Etosha ecosystem, and had more tortuous movements in the wetland environments of Chobe (Table 5; dry season Watson's test statistic: 0.217, $p < 0.05$; wet season Watson's test statistic: 0.200, $p < 0.05$; S2 Appendix). Contrarily, hyenas exhibited more directional movements in Chobe and more tortuous movements in Etosha during dusk and dawn periods of the wet season (Table 5; Watson's test statistic: 0.171, $0.05 < p < 0.10$; S2 Appendix). Similarly, lions had more directional movements in Etosha than they did in Botswana during the nocturnal dry season (Table 5; Watson's test statistic: 0.177, $0.05 < p < 0.10$; S2 Appendix). In addition, lions were also more directional than hyenas over the 24-hour period in Etosha (Table 5; Watson's test statistic: 0.305, $p < 0.01$), and in Chobe during the wet season (Table 5; Watson's test statistic: 0.189, $p < 0.05$; S2 Appendix). However, lions were more tortuous than hyenas were in Etosha during dusk/dawn periods (Table 5; Watson's test statistic: 0.305, $p < 0.01$; S2 Appendix), while in Chobe, hyenas had more tortuous movements than lions during the late morning (S13 Fig, Watson's test statistic 0.269, $p < 0.05$; S2 Appendix).

Moreover, lion and hyena movements and activity differed according to the lunar cycle. During nocturnal periods of the dry season in Etosha, lions had higher activity, travelled further and had more directional movements during periods of low light conditions (i.e., waxing and waning crescents; Table 6, activity $F = 11.33$, step length $F = 16.36$, tortuosity $F = 34.10$, $p < 0.0001$; S2 Appendix). Etosha lions also demonstrated more tortuous movements on new

**Table 6. Lion and spotted hyena activity and movement metrics in relation to the lunar cycle.**

| Period | Region | Species | Metric | | Lunar phase | | | | | | | |
|---|---|---|---|---|---|---|---|---|---|---|---|---|
| | | | | | New moon | Waxing crescent | First quarter | Waxing gibbous | Full moon | Waning gibbous | Last quarter | Waning crescent |
| Nocturnal combined seasons | CNP | Lion | Activity | | 29.34 ± 45.53 | | | | 27.16 ± 44.46 | | | |
| | | | Step length | | 242.81 ± 414.56 | | | | 222.71 ± 394.73 | | | |
| | | | Tortuosity | | *-0.272 ± 2.26* | | | | *0.442 ± 2.58* | | | |
| Nocturnal dry season | ENP | Lion | Activity | | 38.46 ± 52.31 | 45.35 ± 53.94 | 27.17 ± 48.32 | 23.75 ± 45.48 | 33.23 ± 50.35 | 36.16 ± 50.15 | 37.39 ± 52.07 | 37.22 ± 53.67 |
| | | | Step length | | 304.10 ± 453.23 | 400.64 ± 472.46 | 288.77 ± 437.63 | 243.85 ± 400.09 | 327.22 ± 456.94 | 330.55 ± 430.07 | 334.28 ± 449.32 | 389.16 ± 479.82 |
| | | | Tortuosity | | *0.351 ± 1.93* | 0.014 ± 1.82 | -0.170 ± 2.03 | -0.245 ± 2.05 | *0.013 ± 1.88* | 0.164 ± 1.82 | -0.223 ± 1.80 | -0.009 ± 2.00 |
| | ENP | Spotted hyena | Activity | | 78.11 ± 77.72 | 73.21 ± 77.41 | 65.85 ± 76.68 | 86.40 ± 80.02 | 85.07 ± 80.38 | 66.01 ± 79.53 | 75.38 ± 77.10 | 73.23 ± 75.08 |
| | CNP | Spotted hyena | Tortuosity | | 0.139 ± 1.88 | -0.029 ± 1.82 | -0.057 ± 1.77 | -0.046 ± 1.85 | 0.020 ± 1.92 | 0.026 ± 1.89 | 0.075 ± 1.93 | 0.021 ± 1.94 |
| Nocturnal wet season | ENP | Spotted hyena | Tortuosity | | -0.089 ± 1.77 | -0.043 ± 1.60 | 0.084 ± 1.61 | 0.009 ± 1.71 | 0.010 ± 1.65 | 0.028 ± 1.67 | 0.137 ± 1.74 | 0.059 ± 1.74 |
| Dusk/dawn wet season | CNP | Lion | Tortuosity | | 0.785 ± 1.92 | 0.862 ± 1.85 | 1.818 ± 1.93 | 0.145 ± 1.76 | 1.232 ± 1.73 | -1.526 ± 1.77 | 0.988 ± 1.81 | -1.314 ± 1.78 |

Additional metrics are available in S8 Table. Activity (AMVs), step length (m), and path tortuosity (radian) of lions and spotted hyenas from the Etosha National Park, Namibia (ENP), and the Chobe National Park, Linyanti Conservancy, and the NG32 concession of the Okavango Delta[a], Botswana (CNP). Values are means ± standard deviations during the nocturnal (30min fixes from 18h00-6h00 and 17h00-8h00) and dusk/dawn (5min fixes from 19h00-21h00 and 4h00-6h00) periods. Values included here consist of significant differences among means, with italicized values a significant difference between means (ANOVA $F$-test $p < 0.05$).

[a]*No spotted hyenas were collared from the Okavango Delta, Botswana.*

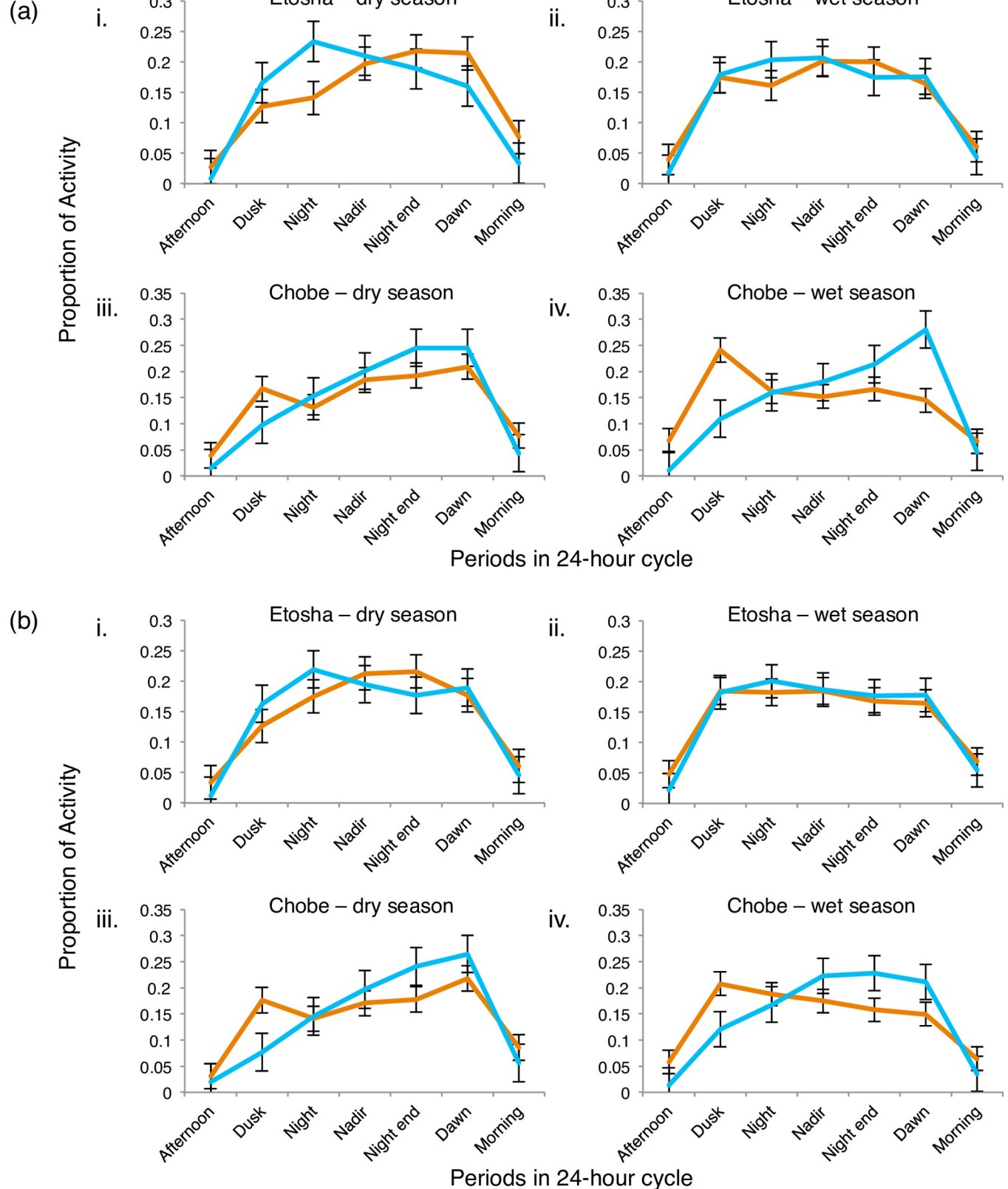

**Fig 6. Seasonal activity of lions and spotted hyenas according to full and new moons.** Plots portraying the proportion of activity of lions (orange lines) and spotted hyenas (blue lines) during full moon (a) and new moon (b) phases for each of the dry and wet seasons in Etosha and Chobe/Linyanti study areas. The

24-hour cycle was subdivided into seven different periods. Night/nadir/night-end consists of the end of evening twilight to the beginning of morning twilight divided into three equal intervals. Afternoon = noon to sundown; dusk = sundown to twilight end; dawn = beginning of morning twilight to sunrise; morning = sunrise to noon. Points represent the mean and error bars the standard error (SE).

moon nights (Table 6, $F = 22.99$, $p < 0.05$). However, hyena activity in Etosha demonstrated two peaks during waxing gibbous to full moon nights, and again during new moon nights, with decreased activity during first quarter phases and after full moon nights (Table 6, $F = 20.13$, $p < 0.0001$). During the wet season, Etosha hyenas had significantly more directional movements during the brightest phases (i.e., waxing/waning gibbous and full moon), and exhibited significantly more tortuous movements during new moon nights, and first and last quarter phases (Table 6, $F = 2.32$, $p < 0.05$).

Similarly, lions in Botswana had higher activity and travelled further on new moon nights (Table 6, activity $F = 19.39$, step length $F = 7.34$; $p < 0.05$), although they had significantly more tortuous movements during full moon nights (Table 6, Watson's test statistic: 0.168, $0.5 < p < 0.10$). In addition, during dusk/dawn periods of the wet season, Botswana lions had mostly tortuous movements in first quarter and waning gibbous, with mostly directional movements in waxing gibbous phases (Table 6, $F = 2.51$, $p < 0.05$). Contrarily, hyenas in Chobe had mostly directional movements during full moon nights and were significantly more tortuous during new moon nights during nocturnal periods of the dry season (Table 6, $F = 4.19$, $p < 0.05$).

Furthermore, the proportion of activity as shown by lions and hyenas according to the lunar cycle during nocturnal periods reveals a seasonal effect on, and regional differences in, their temporal activity patterns. For both the arid and mesic environments, lions and hyenas exhibited significantly higher proportions of activity during the periods of the night between dusk to dawn (Fig 6; Table 7; Lion $t = 9.86$, df = 12.8, $p < 0.0001$; Hyena $t = 20.93$, df = 13.0, $p < 0.0001$), regardless of moon phase. Hyena activity was also significantly higher than lion activity during the period between dusk to dawn (Table 7, $t = 3.47$, df = 11.9, $p < 0.05$).

**Table 7. Proportion of species activity during the nocturnal period.**

**(a)**

| Nocturnal Period | Lion | Spotted hyena |
|---|---|---|
| Night | *0.531 ± 0.014* | *0.591 ± 0.009* |
| Dusk/dawn | *0.356 ± 0.010* | *0.350 ± 0.007* |

**(b)**

| Region | Nocturnal Period | Lion | Spotted hyena |
|---|---|---|---|
| ENP | Dusk-night | **0.159 ± 0.008** | **0.193 ± 0.009** |
| | Nightend-dawn | *0.190 0.009* | 0.177 0.003 |
| CNP | Dusk-night | **0.177 ± 0.013** | **0.129 ± 0.012** |
| | Nightend-dawn | **0.177 ± 0.010** | **0.241 ± 0.008** |

Proportion (mean ± SE) of activity mean values (AMVs) of lions and spotted hyenas during (a) night and dusk/dawn periods, and (b) during initial (dusk-night) and latter (nightend-dawn) periods from each region. AMVs were from the Etosha National Park, Namibia and the Chobe National Park, Linyanti Conservancy and Okavango Delta[a], Botswana. The night period occurs between evening and morning twilight and consists of three equal intervals (night, nadir, and night-end). Values in bold indicate a significant difference between species, and italicized values indicate a significant difference between time periods within species. Greyed out values were not significant.

[a]*No spotted hyenas were collared from the Okavango Delta, Botswana.*

Regardless of the lunar cycle, both Etosha lions and Chobe hyenas demonstrated significantly increased proportions of activity during the latter phase of the night (Fig 6; Table 7; Etosha lion t = -2.56, df = 14.0, $p < 0.05$; Chobe hyena t = -7.88, df = 12.8, $p < 0.05$). In addition, hyenas exhibit heightened proportions of activity in the initial phase of the night when compared to lions (Fig 6; Table 7; $t$ = -2.77, df = 14, $p < 0.05$; S9 Table). However, the opposite is true for Chobe with lions having heightened proportions of activity in the initial phase when compared to hyenas (Fig 6; Table 7, $t$ = -2.81, df = 13.9, $p < 0.05$; S9 Table), with hyenas having higher proportions of activity than lions during the latter phase (Fig 6; Table 7; $t$ = -5.03, df = 13.7, $p < 0.001$; S9 Table). Despite lions and hyenas exhibiting higher proportions of activity during certain periods of the night, there remains a temporal shift in activity between lions and hyenas in which periods of higher proportions of activity is dominated by one species during different time periods of the night. Thus, these distinctive differences in the activity patterns between the two species during the night suggest a temporal partitioning strategy in areas where both species co-exist.

## Inter- and intra-specific effects

**Frequency of time-matched distances.** Of 1,155,488 records of distances between collared individuals that overlapped in time, 46.7% occurred between conspecifics (26.1% for lions and 20.6% for spotted hyenas), and 53.3% occurred between lions and hyenas (competitors). From these data, we extracted all measured records between two collared individuals that occurred at a time-matched distance of $\leq$5 km ($n$ = 34,682). Overall, predators were at time-matched distances of $\leq$5 km to competitors and conspecifics with nearly equal frequency (51% and 49%; $\kappa^2$ = 0.071, df = 1, $p > 0.05$; Fig 7). Etosha lions occurred at time-matched distances of $\leq$5 km with each other significantly more often than Chobe lions did ($\kappa^2$ = 39.69, df = 1, $p < 0.001$), while Chobe hyenas were at time-matched distances of $\leq$5 km with each other more often than Etosha hyenas were ($\kappa^2$ = 8.29, df = 1, $p < 0.005$). Etosha lions occurred significantly more often with competitors at further time-matched distances (1-5 km), and with conspecifics at closer time-matched distances (0-50 m), whereas hyenas occurred at time-matched distances of <5 km with competitors more often than with conspecifics (Fig 7, and S12 Table). However, hyenas in Chobe occurred significantly more often with conspecifics than they did with competitors at time-matched distances of 200 m-1 km. In addition, lions and hyenas tended to be at further time-matched distances to competitors than to conspecifics, although this was not significant in all cases (S12 Table).

**Consecutive time points.** During consecutive time points (which indicates a longer duration of time together), there were significantly more instances of lion-lion dyads at distance intervals of 0-100 m, 100-200 m, 200-500 m, and 500-1000 m than there were of lion-hyena dyads, indicating that lions spend more time together at these distances than they do with hyenas (Fig 8). In addition, intraspecific dyads have more instances (85%) of being together for consecutive time points than interspecific dyads (15%). Lion dyads consistently presented higher values for both the "11-30" and ">30" consecutive time point groups, indicating that they spend more time together at these distances than they did with hyenas, and more than hyenas did (S13 Table). However, despite higher frequencies of hyenas spending more time together than they did with competitors at shorter consecutive intervals (i.e. "1" and "2" groups indicating 5-10 mins), there were more occurrences of lion-hyena dyads at greater consecutive time points (">30" group), indicating a longer time interval of at least 300 consecutive minutes for lion-hyena dyads at distances between 0-2 km.

**Movement and activity within core use areas** The activity recorded from the accelerometers of Chobe lions and hyenas were significantly higher when inside the core use area of the

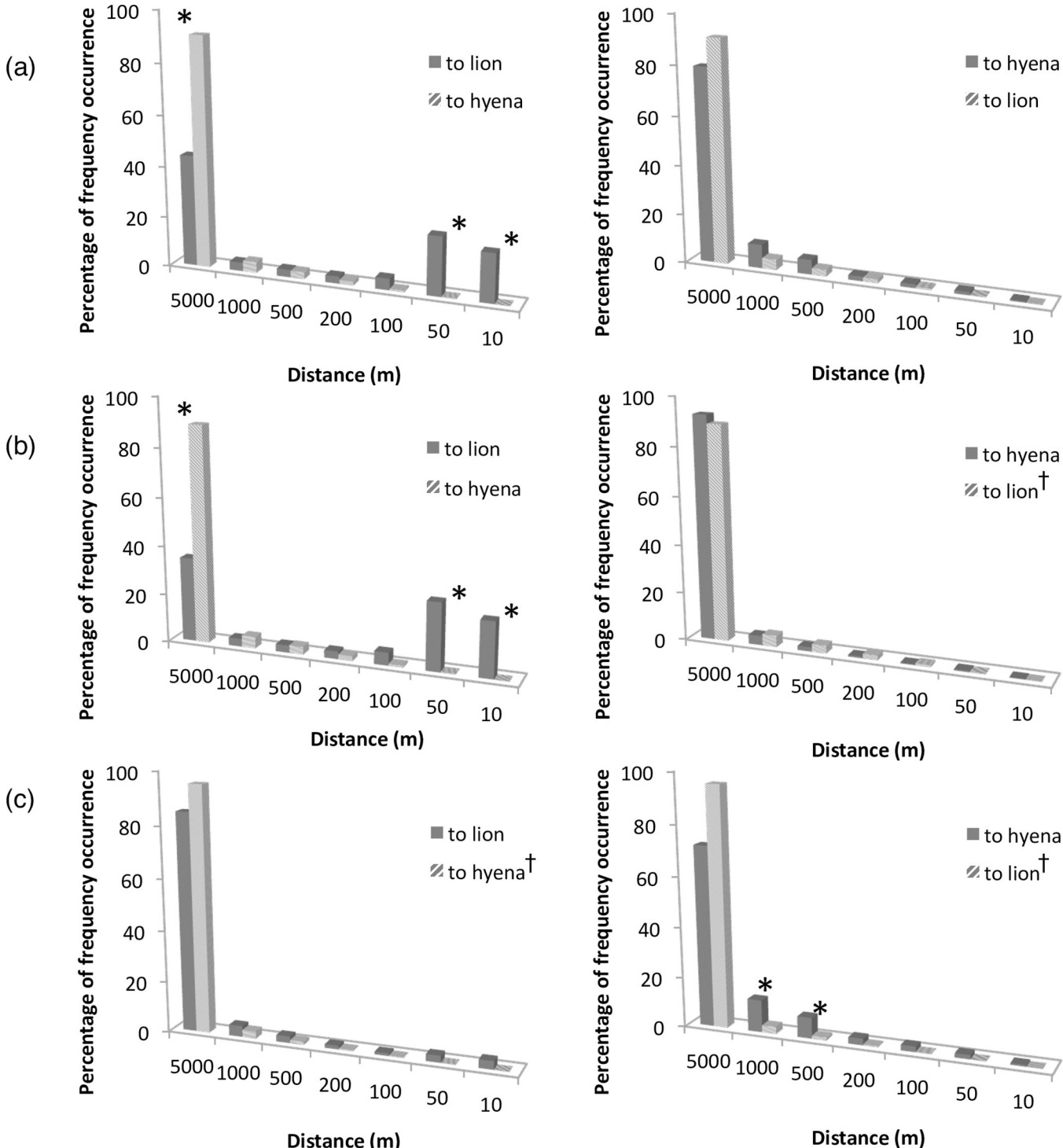

**Fig 7. Frequency of time-matched distances between lions and spotted hyenas.** Percent frequency occurrence of time-matched distances from lions (left side figures) and spotted hyenas (right side figures) to collared individuals at distance intervals of 0-10 m, >10-50 m, >50-100 m, >100-200 m, >200-500 m, >500-1000 m, and >1-5 km. Dark grey bars reflect frequency of distances to conspecifics, and hatched bars frequency of distances to competitors. Plots represent the (a) overall percentage frequency occurrence, (b) Etosha groups, and (c) Chobe/Linyanti groups. Asterisks denotes the significant difference between competing groups for that interval; and a dagger indicates which competing group has the greater percent frequency occurrence between 0-5 km. Statistical results are presented in S12 Table.

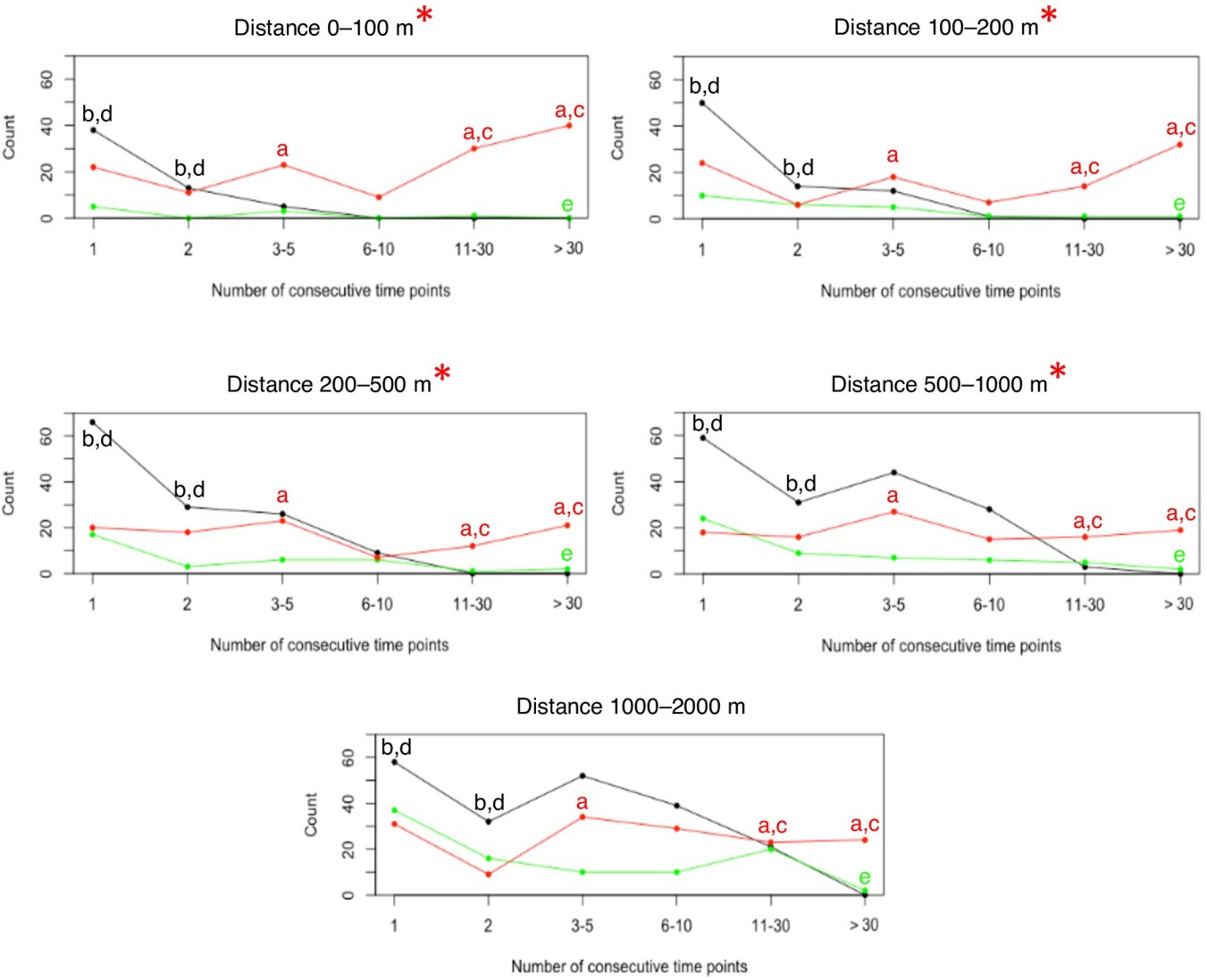

**Fig 8. Consecutive time points between lions and spotted hyenas at different distance intervals.** Number of consecutive time points (indicating longer time duration) for which pairs of lion-lion (red), hyena-hyena (black), and lion-hyena (green) dyads were at distances of 0-100, >100-200, >200-500, >500-1000, or >1000-2000 m. Red asterisks at distance intervals indicate where the lion-lion dyad had significantly more consecutive time points contrasted to lion-hyena dyads. Letters indicate significance for the dyad at that time duration: lion-lion vs lion-hyena (a), hyena-hyena vs lion-hyena (b), lion-lion vs hyena-hyena (c), hyena-hyena vs lion-lion (d), lion-hyena vs hyena-hyena (e). Number of dyads: 9 lion-lion dyads, 4 hyena-hyena dyads, and 10 lion-hyena dyads. Statistical results are presented in S13 Table.

competitor species (Fig 9, S14 Table), although the activity of Etosha lions and hyenas also increased when inside competitor core areas. Chobe lions had higher activity inside competitor core areas during the dusk/dawn period (Table 8, $F = 51.31$, $p < 0.05$), while a hyena from Chobe had higher activity both inside the competitor, and outside the conspecific core areas (Table 8, $F = 111.7$ and 433.1, respectively, all $p$-values < 0.0001).

Additionally, lions in both ecosystems had increased activity during dusk/dawn periods when at closer distances to conspecifics than to competitors (Table 8, Etosha 100-200 m, $t = -3.77$, df = 8.9, $p < 0.01$; Chobe 500-600 m, $t = -80.71$, df = 1, $p < 0.01$), with Etosha lions travelling significantly faster (calculated from GPS locations) at closer distances to conspecifics than to competitors (Table 8, 100-200 m, $t = -6.78$, df = 6.7; 200-300 m, $t = -3.78$, df = 6.7; 300-

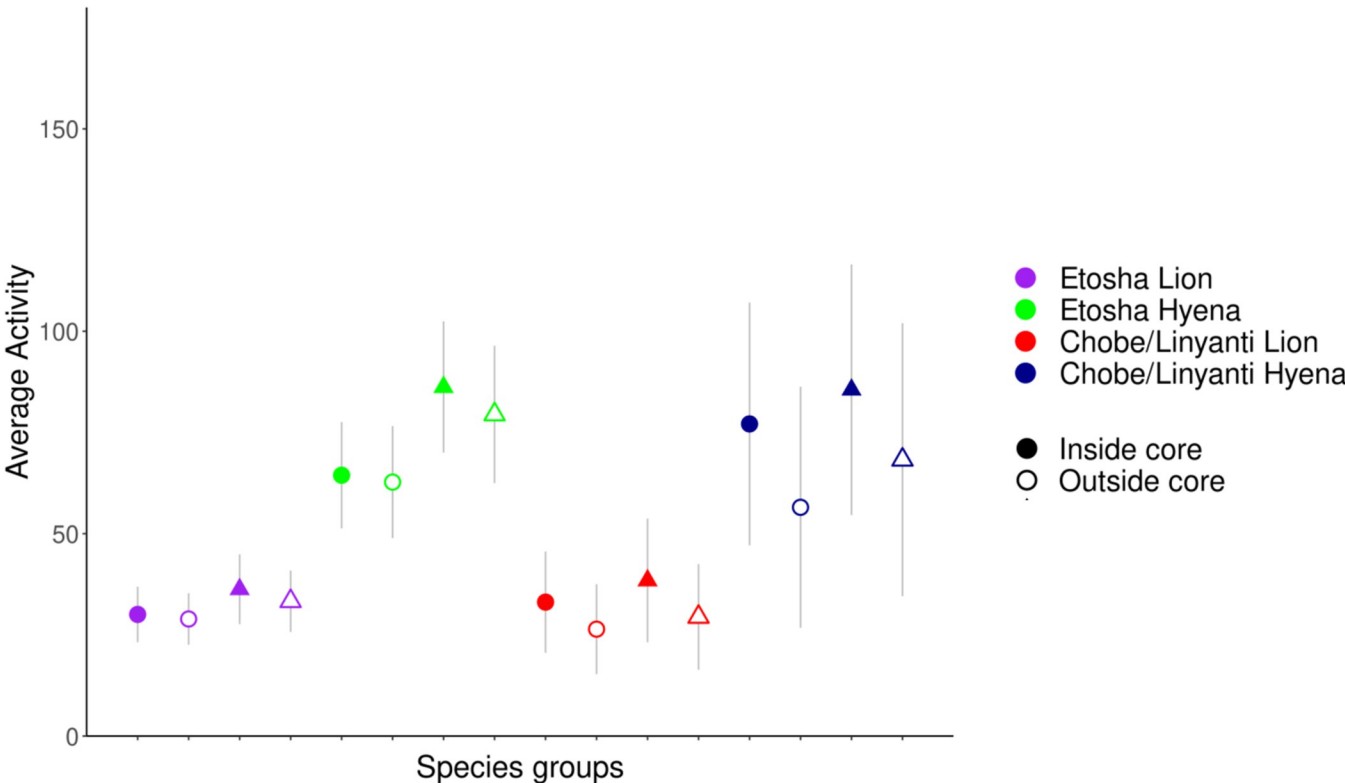

**Fig 9. Average activity of lions and spotted hyenas with respect to competitor core use areas.** Inside competitor core areas are represented by solid shapes, and outside competitor core areas are represented by open shapes. Shapes represent the mean activity and error bars the SE. Activity was measured simultaneously on each axis as the difference in acceleration between two consecutive measurements and given a relative range between 0 and 255 (activity monitor values [AMVs]), characterizing the mean activity/acceleration. Activity X (circles) = forward/backward motion. Activity Y (triangles) = rotary/sideways motion.

400 m, $t$ = -4.96, df = 5.9; 400-500 m, $t$ = -3.25, df = 10.4; all $p$-values < 0.01). Similarly, hyenas from Etosha demonstrated significantly increased activity at closer distances to conspecifics than to competitors during nocturnal periods (Table 8, 100-200 m, $t$ = 2.74, df = 5.0, $p$ < 0.05), whereas they travelled at faster speeds when closer to competitors than to conspecifics during dusk/dawn periods (Table 8, 0-100 m, $t$ = 2.77, df = 5.1, $p$ < 0.05). Contrarily, the dusk/dawn activity of the Chobe hyena was significantly higher at closer distances to competitors than to conspecifics (Table 8, 200-300 m, $t$ = 2.88, df = 3.9; 600-700 m, $t$ = 3.08, df = 15.0; all $p$-values < 0.05).

During dusk/dawn periods, hyenas from both Etosha and Chobe travelled at significantly faster speeds inside of competitor core areas relative to outside of competitor core areas (Table 8, Etosha $F$ = 12.96, $p$ < 0.05; Chobe $F$ = 8.33, $p$ < 0.05). Similarly, hyenas from Chobe also travelled at significantly faster speeds inside of competitor core areas during nocturnal periods (Table 8, $F$ = 18.93, $p$ < 0.05). Conversely, Chobe hyenas travelled at significantly slower speeds when inside of conspecific core areas relative to outside of conspecific core areas during nocturnal periods (Table 8, $F$ = 1515.36, $p$ < 0.05). Additionally, Chobe hyenas moved at significantly faster speeds inside competitor core areas in comparison to when they were inside conspecific core areas during nocturnal (Table 8, $t$ = 6.04, df = 4.2, $p$ < 0.01) and dusk/dawn periods (Table 8, $t$ = 4.88, df = 4.9, $p$ < 0.01).

The tortuosity of lion movements differed when they were inside of competitor core areas in relation to when they were outside of it (Fig 10). During nocturnal periods, lions from

**Table 8. Movement and activity metrics within competitor and conspecific core use areas.**

**(a)**

| Metric | Period | Region | Species | Competitor | | Conspecific | |
|---|---|---|---|---|---|---|---|
| | | | | Inside core | Outside core | Inside core | Outside core |
| Activity | Dusk/dawn | CNP | Lion | 34.29 ± 50.88 | 30.66 ± 43.43 | 33.52 ± 41.75 | 37.16 ± 46.29 |
| | | CNP | Spotted hyena | 84.32 ± 61.43 | 65.34 ± 64.85 | 50.97 ± 56.15 | 77.58 ± 67.21 |
| Speed | Nocturnal | CNP | Spotted hyena | *0.438 ± 0.36* | 0.297 ± 0.36 | *0.233 ± 0.32* | 0.334 ± 0.38 |
| | Dusk/dawn | ENP | Spotted hyena | 0.470 ± 0.49 | 0.422 ± 0.52 | 0.478 ± 0.53 | 0.415 ± 0.51 |
| | | CNP | Spotted hyena | *0.527 ± 0.49* | 0.363 ± 0.47 | *0.268 ± 0.41* | 0.407 ± 0.50 |
| Tortuosity | Nocturnal | ENP | Lion | * 0.055 ± 1.91 | 0.059 ± 1.90 | -0.056 ± 1.99 | 0.138 ± 1.87 |
| | | ENP | Spotted hyena | -0.016 ± 1.57 | 0.027 ± 1.80 | -0.079 ± 1.59 | 0.018 ± 1.74 |
| | | CNP | Lion | * 0.032 ± 2.41 | -0.222 ± 2.43 | *0.113 ± 2.88* | -0.123 ± 2.33 |
| | Dusk/dawn | ENP | Lion | 0.073 ± 2.37 | 0.029 ± 2.51 | -0.369 ± 2.63 | 0.088 ± 2.34 |

**(b)**

| Region | Period | Distance (m) | Species | Metric | Competitor | Conspecific |
|---|---|---|---|---|---|---|
| ENP | Nocturnal | 0-100 | Lion | Tortuosity | -2.912 ± 1.66 | 0.169 ± 2.07 |
| | | 100-200 | Lion | Tortuosity | 2.775 ± 1.58 | 0.170 ± 1.12 |
| | | | Spotted hyena | Activity | 79.20 ± 56.20 | 127.18 ± 31.86 |
| | | 400-500 | Lion | Tortuosity | 2.305 ± 1.27 | -0.001 ± 1.21 |
| | Dusk/dawn | 0-100 | Spotted hyena | Speed | 0.135 ± 0.15 | 0.010 ± 0.01 |
| | | 100-200 | Lion | Activity | 23.32 ± 50.22 | 53.66 ± 49.73 |
| | | | | Speed | 0.082 ± 0.24 | 0.412 ± 0.34 |
| | | | | Tortuosity | 3.130 ± 1.73 | 0.076 ± 1.14 |
| | | 200-300 | Lion | Speed | 0.155 ± 0.26 | 0.337 ± 0.34 |
| | | 300-400 | | | 0.085 ± 0.21 | 0.302 ± 0.39 |
| | | 400-500 | | | 0.135 ± 0.26 | 0.396 ± 0.37 |
| | | | | Tortuosity | -2.666 ± 1.88 | 0.058 ± 1.48 |
| CNP | Dusk/dawn | 200-300 | Spotted hyena | Activity | 92.50 ± 20.07 | 53.06 ± 47.97 |
| | | 500-600 | Lion | Activity | 29.50 ± 0.10 | 45.55 ± 52.24 |
| | | 600-700 | Spotted hyena | Activity | 106.17 ± 27.62 | 59.17 ± 56.41 |

Additional metrics are available in S14 Table. Nocturnal (30min fixes) and dusk/dawn (5min fixes) periods of lion and spotted hyena activity (AMVs), speed (m/s), and path tortuosity (radian) from (a) inside and outside of competitor and conspecific core use areas, and (b) at distance intervals in meters to the nearest competitor and conspecific. Collared individuals were from the Etosha National Park, Namibia (ENP), the Chobe National Park and Linyanti Conservancy, Botswana (CNP). Values consist of significant differences between inside and outside core use areas (a), or between competitors and conspecifics at distance intervals (b). Italicized values indicate significant differences inside core use areas between competitors and conspecifics (a), and values with an asterisk indicate a significant difference inside the competitor core use area between ENP and CNP (a). Greyed out values were not significantly different.

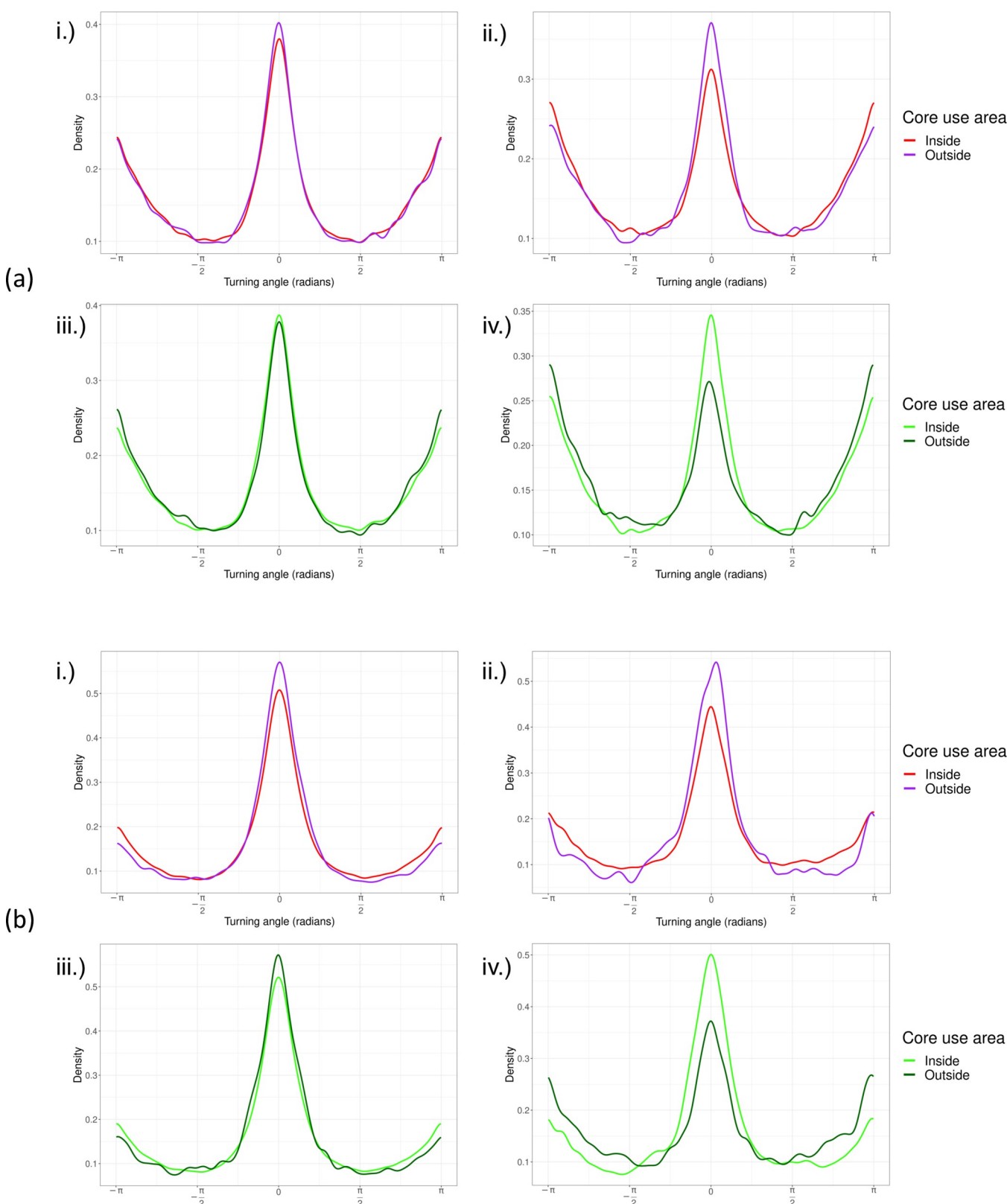

**Fig 10. Tortuosity of lions and spotted hyenas with respect to competitor core use areas.** Tortuosity of (a) lions and (b) spotted hyenas from the Etosha National Park, Namibia (i & iii) and the Chobe National Park and Linyanti Conservancy, Botswana (ii & iv). Path tortuosity is shown from inside and outside of competitor core areas (i & ii), and conspecific core areas (iii & iv).

Chobe had significantly more directional movements inside competitor core areas (Table 8, Watson's test statistic: 0.242, $p < 0.05$), and were also more directional inside competitor core areas than inside of conspecific core areas (Table 8, Watson's test statistic: 0.153, $0.05 < p < 0.10$). In addition, Chobe lions were significantly less tortuous inside competitor core areas than Etosha lions were (Table 8, Watson's test statistic 0.204, $p < 0.05$). Contrarily, Etosha lions had significantly more tortuous movements inside competitor core areas than outside of them during dusk/dawn periods (Table 8, Watson's test statistic: 0.171, $0.05 < p < 0.10$).

Conversely, hyenas from Etosha had significantly more tortuous paths inside conspecific core areas with more directional movements outside conspecific core areas during nocturnal periods (Fig 10; Table 8, $F = 8.65$, $p < 0.05$). Furthermore, lions from Etosha had significantly more tortuous movements when at close distances to competitors (up to 500 m) during nocturnal and dusk/dawn periods, relative to when at close distances to conspecifics (Table 8, nocturnal, 0-100 m Watson's test statistic 0.242; 100-200 m Watson's test statistic 0.227; 400-500 m Watson's test statistic 0.227; dusk/dawn, 100-200 m Watson's test statistic 0.264; 400-500 m Watson's test statistic 0.191; all $p$-values $< 0.05$; S15 Fig).

## Time-use metrics

Revisitation and duration (RD) space plots that indicate areas of high revisitation rates and locations of long visit durations for a lion and spotted hyena are presented in Fig 2 (with all other individuals in S16 and S17 Figs). The distributions of the selected variables for each of the clusters are presented alongside a map of the individual's relocations, color-coded according to the identified clusters within the RD space in Fig 3 (with all other individuals in S18 and S19 Figs). For all individuals, the factor analysis of mixed data (FAMD) method consistently selected land cover categories and time of day as high-scoring variables among the principal dimensions explaining the patterns of recursions and extended stays for each lion and hyena individual. The next most consistently selected ecogeographical variables as important factors explaining the patterns of recursions and extended stays for lions were variables related to interspecific and intraspecific interactions (probability of, distance to, and whether inside of core), chosen 75% and 68.8% of the time, respectively (Table 9). However for hyenas, the most consistently selected factors among the principal dimensions were variables related to interspecific interactions (probability of, distance to, and whether inside of core), chosen 61.5% of the time. Other variables selected with equal consistency as variables of interspecific interactions for Etosha hyenas included probability of site-attracted foragers and moon illumination, while distance to permanent water was selected with equal frequency for Chobe hyenas (Table 9).

Focusing our time-use observations on variables associated with interspecific competition, we observed that lions tended to demonstrate longer durations in locations of low competitor probabilities, at far distances to competitors, and when outside competitor core use areas (see a.3 and a.8 in S18 Fig). Lions also had increased recursions in areas of high competitor probabilities, at close distances to competitors and inside competitor core use areas (all chi-square $p$-values $< 0.001$; Table 10). However, male lions and females of mating pairs exhibited longer durations in localities of high competitor probabilities; while single female lions had shorter durations with increased revisitations (see a.5 and c.3 versus a.4 and b.2 in S18 Fig). Similarly, hyenas also tended to have longer durations in locations of low competitor probabilities, and they had higher recursions with short durations in locales of high competitor probabilities (see a.1 and b.3 in S19 Fig). Furthermore, hyenas exhibited longer durations at further distances from competitors with shorter durations and increased revisitations at shorter distances to competitors (all chi-square $p$-values $< 0.001$; Table 10; see a.6 and c.2 in S19 Fig).

**Table 9. Highest-scoring ecogeographical variables for lions and spotted hyenas.**

| Ecogeographical Variables | | Lion | | | Spotted hyena | | |
|---|---|---|---|---|---|---|---|
| | | Etosha | Chobe/Linyanti | Overall | Etosha | Chobe/Linyanti | Overall |
| | | *n* = 10 | *n* = 6 | *n* = 16 | *n* = 8 | *n* = 5 | *n* = 13 |
| *Bioclimatic variables* | Time of day | 100 | 100 | 100 | 100 | 100 | 100 |
| | Temperature | 40 | 33.3 | 37.5 | 50 | 20 | 38.5 |
| | Season | 20 | 33.3 | 25 | 12.5 | 40 | 23.1 |
| | Moon Illumination | 20 | 16.7 | 18.8 | 62.5 | 20 | 46.2 |
| | Precipitation amounts | 40 | 0 | 25 | 25 | 0 | 15.4 |
| *Landscape features* | Slope | 50 | 16.7 | 37.5 | 50 | 40 | 46.2 |
| | Land cover category | 100 | 100 | 100 | 100 | 100 | 100 |
| | NDVI | 20 | 16.7 | 18.8 | 12.5 | 20 | 15.4 |
| | Probability of site-attracted foragers | 40 | - | - | 62.5 | - | - |
| | Distance to carcasses | 30 | - | - | 37.5 | - | - |
| | Distance to permanent water | - | 50 | - | - | 60 | - |
| | Distance to seasonal water | 30 | 16.7 | 25 | 37.5 | 20 | 30.8 |
| | Distance to roads | 0 | 0 | 0 | 12.5 | 0 | 7.7 |
| *Human disturbances* | Distance to anthropogenic features | 40 | 0 | 25 | 0 | 40 | 15.4 |
| *Competitive effects* | Interspecific interactions | 80 | 66.7 | 75 | 62.5 | 60 | 61.5 |
| | Intraspecific interactions | 70 | 66.7 | 68.8 | 37.5 | 40 | 38.5 |

Frequency percentage occurrence of highest-scoring variables among the principal dimensions as important factors to explain >80% of the cumulative variation for each of Etosha, Chobe/Linyanti and overall lions and spotted hyenas, as selected by the FAMD (factor analysis of mixed data) method.

However, with regards to intraspecific competition, our observations revealed no differences in the time-use duration and revisitation of lions in locations of low and high conspecific probabilities (Table 10). Conversely, hyenas had twice the frequency of extended durations than they did shorter durations in locations of higher conspecific probabilities (chi-square *p*-value < 0.05; Table 10; see a.2 and a.3 in S19 Fig). In addition, hyenas also had higher recursions in localities of lower conspecific probabilities, at far distance to conspecifics, and outside conspecific core use areas (chi-square *p*-value < 0.001; Table 10).

## Discussion

Combining patterns of space use, temporal activity, fine-scale habitat use differentiation, and localized reactive avoidance behaviors in response to the potential risk of competition, has revealed the complex dynamics among lions and spotted hyenas within an apex predator system. Our findings are consistent with other studies in which lions and hyenas are often positively associated with one another [8, 31, 42, 75], and they appear to behaviorally mediate the potential for competition by active avoidance or fine-scale behavioral mechanisms, as exhibited by other carnivores in response to the direct risk of encountering other predators [30, 92].

Environmental spatial complexity, which allows for the selection of different habitats, may promote coexistence between species [93], and has been recorded for several sympatric carnivores [15, 24, 37, 94–100], including lions and spotted hyenas [30, 101]. However, fine-scale habitat use differentiation likely occurs as the key mechanism allowing for the coexistence of

**Table 10. Frequency occurrence of revisitation and duration for lions and spotted hyenas according to social interactive covariates.**

| Species | Interactive variables | Revisitation | Duration | | Significance |
|---|---|---|---|---|---|
| Lion | • - high competitor probability | 73% | 27% | $\kappa^2 = 21.2$ | *** |
| | • - close distance to competitor(s) | | | $df = 1$ | |
| | • - inside competitor core use area | | | $p < 0.001$ | |
| | • - low competitor probability | 27% | 73% | $\kappa^2 = 21.2$ | *** |
| | | | | $df = 1$ | |
| | • - far distance to competitor(s) | | | $p < 0.001$ | |
| | • - outside competitor core use area | | | | |
| | • - high conspecific probability | 50% | 50% | $\kappa^2 = 0$ | |
| | - close distance to conspecific(s) | | | $df = 1$ | |
| | • - inside conspecific core use area | | | $p > 0.05$ | |
| | • - low conspecific probability | 50% | 50% | $\kappa^2 = 0$ | |
| | • - far distance to conspecific(s) | | | $df = 1$ | |
| | • - outside conspecific core use area | | | $p > 0.05$ | |
| Spotted hyena | • - high competitor probability | 83% | (17%) | $\kappa^2 = 43.6$ | *** |
| | • - close distance to competitor(s) | | | $df = 1$ | |
| | • - inside competitor core use area | | | $p < 0.001$ | |
| | • - low competitor probability | 0% | 100% | $\kappa^2 = 100$ | *** |
| | • - far distance to competitor(s) | | | $df = 1$ | |
| | • - outside competitor core use area | | | $p < 0.001$ | |
| | • - high conspecific probability | 40% | 40% (20%) | $\kappa^2 = 8$ | * |
| | | | | $df = 2$ | |
| | • - close distance to conspecific(s) | | | $p < 0.05$ | |
| | • - inside conspecific core use area | | | | |
| | • - low conspecific probability | 60% | 20% (20%) | $\kappa^2 = 32$ | *** |
| | • - far distance to conspecific(s) | | | $df = 2$ | |
| | • - outside conspecific core use area | | | $p < 0.001$ | |

Percent frequency occurrence of revisitation (nsv, number of separate visits) and extended duration (mnlv, mean number of locations per visit) obtained from the *k*-prototype clustering of social interactive covariates in the RD space of each individual. Significance from chi-square analyses is denoted with an asterisk at the alpha level of 0.05, and with a triple asterisk at 0.001. Numbers in brackets indicate short durations.

species within homogeneous landscapes [99, 102]. Therefore, while the heterogeneous landscapes of the Chobe riverfront may facilitate coexistence of species of the same trophic level [103], subtle patterns of habitat use partitioning reflected in the movement decisions of lions and hyenas may further explain the persistence of the two predators across arid and mesic extremes of their environment. Our results indicate differences between lion and hyena movements, which can be attributed to differences in the hunting strategies of the two species [104], potentially resulting in fine scale habitat separation rather than complete segregation [105].

As mainly cursorial predators, spotted hyenas are generally more active, travel faster and further than lions [38, 53]. As sit-and-wait predators, lions typically have lower activity, move at slower speeds, and cover shorter distances than hyenas [40, 59]. Corresponding to the findings of Durant et al. [77], lions and hyenas in this study utilized similar habitats, both mainly occurring within grassland habitats in Etosha and shrubland habitats in Botswana [106–108]. Despite utilizing similar habitats, how the two species exploit these habitats differs, and reflects the differences in habitat characteristics between ambush and cursorial predators [109, 110].

Lions typically use areas in which stalking cover is available [58, 111], whereas hyenas prefer to use open areas, and are generally more dispersed across the landscape [112, 113]. Likewise, Bender et al. [16] reported fine-scale habitat segregation among pumas (*Puma concolor*), coyotes (*Canis latrans*) and bobcats (*Lynx rufus*) in the San Andres Mountains as a result of preferences for habitat characteristics that facilitate movements, despite being positively associated with one another. Thus, our results lends support to a growing body of evidence that demonstrates coexistence among carnivores is facilitated by behavioral mechanisms, in addition to spatial and temporal partitioning [16, 32, 51, 114–116].

Although the movement decisions and behavioral responses of lions and hyenas are adaptable across different systems, we found that lions and hyenas do not respond to inter- and intraspecific interactions equally among heterogeneous and homogeneous environments. Hyena clans in Etosha were observed to participate in more territorial clashes than they did in Chobe (unpublished data). As heterogeneous environments, or habitats of increased complexity, permit coexistence among carnivore species [5, 117], presumably the potential for competition (both inter- and intraspecific) is mitigated in Chobe [15], an environment of increased complexity [118].

Previous studies have demonstrated lion movements to be inextricably tied to the location of water-holes across the landscape due to concentrated search efforts for prey [26, 119]. In Etosha during the dry season, water is primarily supplied at developed locations from a system of pumped boreholes that are routinely serviced and maintained by park personnel. Conversely, perennial water from the Chobe river and the Okavango Delta provides a permanent water source at the Botswana sites [120]. In this study, lion core use areas either encompassed, or were anchored, by such water sources, while only half or fewer of hyena core use areas were. Our results indicate that hyenas in Chobe spent more time in areas that were closer to permanent water and had increased recursion to locations further from water. Spotted hyenas have been found to use locations far from water for den sites and resting [31, 114], thus we suggest that hyenas are potentially choosing to remain in areas with landscape characteristics that minimizes detection [121] while increasing prey vulnerability [122]. Although hyenas require access to drinking water, they can survive on very little of it [39]. Presumably, hyenas in Chobe spend more time in riverine habitats which consist of relatively dense vegetative cover and greater topographic heterogeneity to increase the potential of obtaining food resources when prey aggregate at water sources [60, 123]. In areas where lions are the dominant predator, spotted hyenas may be relegated to suboptimal areas away from readily-available and prime resources, similar to coyotes and wolves (*Canis lupus*) [124]. In this way, hyenas are potentially equivalent to naïve or subordinate lions that have been forced into peripheral habitats [125]. Thus, the behavioral plasticity and opportunistic behavior in foraging and habitat use of hyenas may facilitate coexistence with lions in areas where they overlap.

Interestingly, hyenas in Etosha had higher recursions to locations with higher probabilities of encountering site-attracted foragers, thus increasing their chances of coming across foraging ungulates or potentially new carcasses [126]. However, hyenas that shifted their ranges in the wet season to include the anthrax endemic areas had longer durations at these locations; presumably because they had to travel further (±60 km) to access these localities. The ability to leave their home range and traverse across territories of other clans to benefit from an abundance of resources (i.e. surplus carcasses from anthrax outbreaks), is reminiscent of the commuting behavior exhibited by spotted hyenas as they follow to exploit migratory prey during the wet season in the Serengeti [55, 127]. We suggest this behavioral plasticity is an ecological strategy that hyenas appear to exploit as necessary to increase the potential appropriation of resources, thus conferring a fitness advantage.

Furthermore, recursion and duration were synonymous with travelling movements for Etosha hyenas during new moon and full moon periods. Hyena movements were more localized at higher durations during the periods between new and full moons as they spent more time foraging or searching for prey. Similar to wolves, which were documented to be nearly twice as successful when hunting on moonlit nights [128], hyenas likely require sufficient visibility to increase their hunting success [14]. During this study, hyenas were observed to undertake cursorial hunts during moonlit nights and switched to an ambush strategy coupled with opportunistic chases during darker periods. Comparably, lions typically experience higher hunting success during dark nights, although they were less successful at appropriating prey during moonlit nights [129]. In addition, hyenas were observed (during night follows in this study) to rest when the moon was at its brightest during full moon nights and would only resume foraging activities after the moon had lowered in the sky (pers. obs.). This suggests that hyenas are likely to focus on directed movements (traversing between patches) during dark periods, and resting when it is too bright to avoid detection by prey [130]. Accordingly, hyenas focus their foraging and hunting efforts during periods of sufficient light conditions between the time of new and full moon nights, allowing for avoidance of the potential risk of interspecific encounters with lions [131, 132].

Temporal heterogeneity in conjunction with spatial heterogeneity likely occurs as a mitigation strategy in response to interference competition, and facilitates coexistence among carnivores [37, 133]. Our findings indicate lions and hyenas were both nocturnal, with the lion more diurnal than the hyena. Similarly, Sogbohossou et al. [101] found the activity of lions and hyenas to be spread over the night with no real peaks. However, we found evidence of temporal partitioning on a finer scale than nocturnal and crepuscular patterns, as has been recorded in other studies [14, 27, 32, 134]. Thus, our results correspond with studies in which temporal partitioning occurred as a result of differences in activity periods between predator species, and was presumed to be the main driver for coexistence in sympatric carnivores [3, 9, 95].

## Intraspecific and interspecific interactions

Since lions tend to exhibit a high degree of coordinated movements among lion pairs [135], intraspecific interactions were also selected as important factors accounting for the time-use patterns among lions. However, as we did not differentiate between competitive or mutually beneficial intraspecific interactions, it is plausible that mating pairs may have exaggerated or confounded this result. Furthermore, interspecific interactions were important influences on the time-use metrics that drives both lion and hyena space-use patterns. Surprisingly, although female lions had shorter durations in locations with high competitor probabilities, we found that male lions (and their mating partners) had longer durations in these areas. As dominant predators, male lions are either unaffected by being in locations of higher competitor probabilities, or it is likely that they spend more time within these areas to increase the likelihood of encountering hyenas, because they appear to derive benefits from hyenas through kleptoparasitism [44, 136–139]. During this study, male lions were often observed to be attracted to spotted hyena calls: they would immediately perk up their ears, turn to face the direction of the calls, and often walked towards the direction of the hyena sounds. Observations of direct interactions between lions and hyenas during this study almost always involved hyenas actively avoiding male lions by retreating and moving away, as was also observed in other studies [44, 136]. At encounters with male lions at fresh kills, hyenas often lost their kills to lions or aggregated into large groups (>20 individuals) before attempting to initiate mobbing behavior, similar to the findings of previous studies [106, 140].

In addition, our results demonstrated patterns of recursions and locations of extended stays within the home ranges of apex predators to be influenced by the probability of, and proximity to, competitor and conspecifics. We suggest this to be the result of a perceived risk of competition, in which carnivores behaviorally mediate the potential for competition by altering their space use, or movement patterns [28]. Furthermore, other studies have documented the behavioral response of carnivore species to the potential risk of either encountering competitors or competitive interactions with increased vigilance and movements, either in preparation for potential interactions, or to move through/exit the area as quickly as possible [30, 50, 141]. Pangle and Holekamp [142] attributed the vigilance levels of spotted hyenas to be more influenced by interspecific than intraspecific threats. Likewise, Valeix et al. [143] documented lions moving at quicker speeds, and with relatively straighter trajectories, in response to the risk of conflict when close to human settlements. Specifically, we found spotted hyenas to behaviorally reduce the risk of potential interaction with lions by remaining longer in areas of low competitor probabilities and at far distances to competitors. This type of behavior is mirrored in the behavior-specific habitat selection of lions' in response to mitigating the potential risk of conflict with humans [144]. Thus, the behavioral responses of lions and hyenas towards the perceived risk of competition appears to be similar, regardless of whether it stems from inter- or intraspecific interactions.

## Conclusions

Our results have implications for the conservation of large carnivores in substantiating the potential effects of interference competition on lion and spotted hyena spatial patterns and movements. These findings supplement the growing body of evidence that demonstrates coexistence among carnivores is facilitated by fine-scale behavioral mechanisms in addition to spatial and temporal partitioning. While the patterns of spatial and temporal overlap observed among lions and hyenas do not differ from earlier studies; combining patterns of space use, temporal activity, fine-scale habitat use differentiation, and localized reactive avoidance behaviors in response to the potential risk of competition, has revealed the complex dynamics among lions and hyenas within an apex predator system. Additionally, patterns of recursion and locations of extended stays within the home ranges of apex predators are influenced by the probability of and proximity to competitors and conspecifics, and can be used to inform management strategies for the maintenance of carnivore communities.

As large carnivores are becoming increasingly constrained to protected areas, it is important to note that lions and hyenas do not respond to inter- and intraspecific interactions equally among heterogeneous and homogeneous environments. Specifically, the movement decisions and behavioral responses of lions and hyenas are adaptable across different systems, and are likely a result of several factors, including habitat complexity, hunting strategies, and the active avoidance of interspecific and heterospecific competitors. Consequently, we encourage conservation practitioners to recognize the importance of the potential effects of inter- and intraspecific interactions among apex predators in managing diverse, ecological communities.

## Supporting information

**S1 Appendix. Additional details on study sites and data collection.**
(PDF)

**S2 Appendix. Descriptive analyses of lion and spotted hyena movement descriptors.**
(PDF)

**S1 Table. Percent frequency of lion and spotted hyena relocations by land cover type.** Relocations were recorded from lions and spotted hyenas in the Etosha National Park, Namibia, the Chobe National Park and Linyanti Conservancy, Botswana, and only from lions in the NG32 concession of the Okavango Delta, Botswana.
(PDF)

**S2 Table. Seasonal utilization distributions (UDs) for all collared lion and spotted hyena individuals.** Kernel density (a, c) and $a$-LoCoH (b, d) area measures for home ranges and core use areas (km$^2$) of lion (a, b) and spotted hyena (c, d) individuals. Upper panels consists of lion and hyena individuals from the Etosha National Park, Namibia, with lower panels from the Chobe National Park and Linyanti Conservancy. For lions, the NG32 concession in the Okavango Delta, Botswana in also included. CS = combined seasons, DS = dry season, WS = wet season.
(PDF)

**S3 Table. Areas of overlap in lion and spotted hyena ranges.** Total overlapped areas in km$^2$ between lions' (vertical column) and spotted hyenas' (horizontal column) home ranges and core areas in the (a) Etosha National Park, Namibia; (b) Chobe National Park and Linyanti Conservancy, Botswana. Utilization distributions were generated with the home range (95%) and core use area (50%) kernel density estimator (i) and $a$-LoCoH (ii) isopleths. Males are underlined. An asterisk denotes mortality.
(PDF)

**S4 Table. Proportion of overlaps with spotted hyenas in lion ranges.** Total proportion of lion (vertical column) home ranges and core areas overlapped by spotted hyena individuals (horizontal column) in the (a) Etosha National Park, Namibia; (b) Chobe National Park and Linyanti Conservancy, Botswana. Utilization distributions were generated with the home range (95%) and core use area (50%) kernel density estimator (i) and $a$-LoCoH (ii) isopleths. Males are underlined. An asterisk denotes mortality.
(PDF)

**S5 Table. Proportion of overlaps with lions in spotted hyena ranges.** Total proportion of spotted hyena (vertical column) home ranges and core areas overlapped by lion individuals (horizontal column) in the (a) Etosha National Park, Namibia; (b) Chobe National Park and Linyanti Conservancy, Botswana. Utilization distributions were generated with the home range (95%) and core use area (50%) kernel density estimator (i) and $a$-LoCoH (ii) isopleths. Males are underlined. An asterisk denotes mortality.
(PDF)

**S6 Table. Statistical results of the distances between competitor core areas and home range boundaries.** $T$-tests comparing the distances between the center of individual core areas versus the competitors' home range boundary in the Etosha National Park, Namibia (ENP) and the Chobe National Park and Linyanti Conservancy, Botswana (CNP). An asterisk denotes significance at the alpha level with $^* < 0.05$, and $^{***} < 0.001$.
(PDF)

**S7 Table. Lion and spotted hyena movement metrics.** (a) all lions and spotted hyenas, and (b) lions and spotted hyenas from the Etosha National Park, Namibia (ENP) and the Chobe National Park, Linyanti Conservancy, and the NG32 concession of the Okavango Delta[†], Botswana (CNP). Movement metrics consist of the means ± standard deviations for activity (activity monitor values [AMVs]), step length (m), speed (m/s), net-squared displacement (NSD, km$^2$), and path tortuosity (radian). [†]*No spotted hyenas were collared from the Okavango*

*Delta, Botswana.*
(PDF)

**S8 Table. Lion and spotted hyena activity and movement metrics in relation to the lunar cycle.** Activity (AMVs), step length (m), and path tortuosity (radian) of lions and spotted hyenas from the Etosha National Park, Namibia (ENP), and the Chobe National Park, Linyanti Conservancy, and the NG32 concession of the Okavango Delta[†], Botswana (CNP). Values are means ± standard deviations during the nocturnal (30min fixes from 18h00-6h00 and 17h00-8h00) and dusk/dawn (5min fixes from 19h00-21h00 and 4h00-6h00) periods. [†]*No spotted hyenas were collared from the Okavango Delta, Botswana.*
(PDF)

**S9 Table. New and full moon activity of lions and spotted hyenas from the 24-hour period cycle.** Activity (AMVs) of lions and spotted hyenas from the Etosha National Park, Namibia (ENP), and the Chobe National Park, Linyanti Conservancy, and the NG32 concession of the Okavango Delta[†], Botswana (CNP). The 24-hour cycle was subdivided into seven different periods. Night/nadir/night end consists of the end of evening twilight to the beginning of morning twilight divided into three equal intervals. Afternoon = noon to sundown; dusk = sundown to twilight end; dawn = beginning of morning twilight to sunrise; morning = sunrise to noon. Values indicate the means ± standard deviations of the seven different periods of the 24-hour cycle during new and full moon phases of the dry and wet seasons. [†]*No spotted hyenas were collared from the Okavango Delta, Botswana.*
(PDF)

**S10 Table. Lion and spotted hyena movement metrics according to low, medium, and high body condition scores.** Speed (m/s) and path tortuosity (radian) of lions and spotted hyenas during nocturnal (30min fixes) and dusk/dawn (5min fixes) periods from the Etosha National Park, Namibia, the Chobe National Park, Linyanti Conservancy, and the NG32 concession of the Okavango Delta[†], Botswana. Values are indicated in means ± standard deviations. Body conditions of individuals were scored from spinal palpations of immobilized individuals during capture events. [†]*No spotted hyenas were collared from the Okavango Delta, Botswana.*
(PDF)

**S11 Table. Lion and spotted hyena movement metrics according to site-attracted foragers in the Etosha National Park, Namibia.** Speed (m/s) and path tortuosity (radian) of lions and spotted hyenas during nocturnal (30min fixes) and dusk/dawn (5min fixes) periods according to the probability of site-attracted foragers (i.e. ungulates) in sites that consisted of anthrax positive carcasses from previous years.
(PDF)

**S12 Table. Statistical results to accompany Fig 7 in the main text.** Chi-square and *t*-test results for the percent frequency occurrence of, and average distance in meters to, the nearest conspecific and competitor for each bin of distance intervals. An asterisk denotes significance at the alpha level with * < 0.05, ** < 0.01, *** < 0.005, and **** < 0.001.
(PDF)

**S13 Table. Statistical results to accompany Fig 8 in the main text.** *T*-tests of consecutive time points among dyads (indicating longer time duration) at various distance intervals (a), and the frequency occurrences of dyads for different consecutive time points (b). An asterisk denotes significance at the alpha level with * < 0.05, ** < 0.01, *** < 0.005, and **** < 0.001.
(PDF)

**S14 Table. Movement and activity metrics within competitor and conspecific core use areas.** Nocturnal (30min fixes) and dusk/dawn (5min fixes) periods of lion and spotted hyena activity (AMVs), speed (m/s), and path tortuosity (radian) from (a) inside and outside of competitor and conspecific core use areas, and (b) at distance intervals in meters to the nearest competitor and conspecific. Collared individuals were from the Etosha National Park, Namibia (ENP), the Chobe National Park and Linyanti Conservancy, Botswana (CNP).
(PDF)

**S1 Fig. Temporal schedule of collar overlap for lions and spotted hyenas.** Individuals of the Etosha National Park, Namibia (left) are separated from individuals of the Chobe National Park, Linyanti Conservancy, and Okavango Delta, Botswana (right) at the bold line indicated on 6 April 2015. Males are denoted with an asterisk.
(PDF)

**S2 Fig. Lion nocturnal space use.** Utilization distributions were constructed with the KDE (far left and second from right) and LoCoH a-method (second from left and far right). Panels represent the 95% isopleth of the individual's home range for the dry season (left two panels) and wet season (right two panels). Unique identifiers are depicted vertically on the left of each row of maps. Maps indicate the individual's home range on a satellite image of the (a) Etosha National Park with the salt pan visible; (b) Chobe National Park with the Chobe river from west to east; (c) Linyanti Conservancy with the Linyanti river from southwest to northeast; and the (d) NG32 concession in the Okavango Delta on the southwestern tip of Chief's Island. Map source: Google Imagery, TerraMetrics.
(PDF)

**S3 Fig. Spotted hyena nocturnal space use.** Utilization distributions were constructed with the KDE (far left and second from right) and LoCoH a-method (second from left and far right). Panels represent the 95% isopleth of the individual's home range for the dry season (left two panels) and wet season (right two panels). Unique identifiers are depicted vertically on the left of each row of maps. Maps indicate the individual's home range on a satellite image of the (a) Etosha National Park with the salt pan visible; (b) Chobe National Park with the Chobe river from west to east; (c) Linyanti Conservancy with the Linyanti river from southwest to northeast. Map source: Google Imagery, TerraMetrics.
(PDF)

**S4 Fig. Seasonal average speed (m/s) of (a) lions and (b) spotted hyenas from the Etosha National Park, Namibia (left panels) and the Chobe National Park, Linyanti Conservancy, and Okavango Delta[†], Botswana (right panels) during nocturnal periods.** Both figures, males upper panels and females lower panels. Dry season = red lines, and wet season = blue lines. Solid lines represent the mean, and shaded bars are 95% CI. [†]*No spotted hyenas were collared from the Okavango Delta, Botswana.*
(PDF)

**S5 Fig. Step lengths of lions and spotted hyenas from the Etosha National Park, Namibia (left panel) and the Chobe National Park, Linyanti Conservancy, and Okavango Delta[†], Botswana (right panel).** Violins depict the probability distribution, with black dots the mean and black lines the 95% confidence intervals. [†]*No spotted hyenas were collared from the Okavango Delta, Botswana.*
(PDF)

**S6 Fig. Mean step length (m/min) of lions and spotted hyena individuals from the Etosha National Park, Namibia.** (a) 24 hour cycle binned into six 4 hour periods. (b) Nocturnal cycle

of 18h00-6h00 binned into hourly periods. Bars represent the mean and error bars the SE, only upper error bars are shown. Lion individuals (n = 11) represented by pink/purple/blue colors. Spotted hyena individuals (n = 8) represented by green/yellow/orange colors.
(PDF)

**S7 Fig. Mean step length (m/min) of lions and spotted hyena individuals from the Chobe National Park, Linyanti Conservancy, and Okavango Delta[†], Botswana.** (a) 24 hour cycle binned into six 4 hour periods. (b) Nocturnal cycle of 17h00-8h00 binned into hourly periods. Bars represent the mean and error bars the SE, only upper error bars are shown. Lion individuals (n = 8) represented by pink/purple/blue/blue-green colors. Spotted hyena individuals (n = 5) represented by green/yellow/orange colors. [†]*No spotted hyenas were collared from the Okavango Delta, Botswana.*
(PDF)

**S8 Fig. Mean step length at dusk (red circles) and dawn (black circles) for lions (a) and spotted hyenas (b) from the Etosha National Park, Namibia and the Chobe National Park and Linyanti Conservancy, Botswana.**
(PDF)

**S9 Fig. Seasonal mean net-squared displacement (km2) over 24 hour cycles for (a) lions and (b) spotted hyenas across each season from the Etosha National Park, Namibia (left panels) and the Chobe National Park, Linyanti Conservancy, and Okavango Delta[†], Botswana (right panels).** Males dark grey boxes, females white boxes. Boxplots show medians, 25% and 75% quartiles. Dashed lines indicate means. Whiskers indicate the IQR range. [†]*No spotted hyenas were collared from the Okavango Delta, Botswana.*
(PDF)

**S10 Fig. Mean net-squared displacement (km2) of (a) lions and (b) spotted hyenas between nocturnal (sunset – sunrise) and diurnal (sunrise – sunset) periods from the Etosha National Park, Namibia (left panels) and the Chobe National Park, Linyanti Conservancy, and Okavango Delta[†], Botswana (right panels).** Both figures, males top panels and females bottom panels. Boxplots show medians, 25% and 75% quartiles. Dashed lines indicate means. Whiskers indicate the IQR range. [†]*No spotted hyenas were collared from the Okavango Delta, Botswana.*
(PDF)

**S11 Fig.** Frequency density of seasonal turning angles for (a) lions and (b) spotted hyenas from the Etosha National Park, Namibia (left panels) and the Chobe National Park, Linyanti Conservancy, and Okavango Delta[†], Botswana (right panels) during nocturnal (sunset – sunrise) and diurnal (sunrise – sunset) cycles. Both figures, males in upper panels and females in lower panels. Dry season = red lines, wet season = blue lines. [†]*No spotted hyenas were collared from the Okavango Delta, Botswana.*
(PDF)

**S12 Fig.** Frequency density of seasonal turning angles for (a) lions and (b) spotted hyenas during nocturnal (18h00-6h00 and 17h00-8h00) and dusk/dawn (19h00-21h00 and 4h00-6h00) periods. Etosha National Park, Namibia, left panels and Chobe National Park, Linyanti Conservancy, and Okavango Delta[†], Botswana, right panels. Both figures, males upper panels and females lower panels. Dry season = red lines, wet season = blue lines. [†]*No spotted hyenas were collared from the Okavango Delta, Botswana.*
(PDF)

**S13 Fig.** Frequency density of seasonal turning angles for (a) lions and (b) spotted hyenas during each hour of the nocturnal period (18h00-6h00) and during 4 hour blocks of the diurnal period (6h00-18h00). Species are from the Etosha National Park, Namibia (dry season = magenta lines; wet season = cyan lines), the Chobe National Park and Linyanti Conservancy, Botswana, with lions from the Okavango Delta, Botswana (dry season = red lines; wet season = blue lines). A double asterisk indicates a significant difference in the Watson's Two-Sample Test of Homogeneity at $p < 0.05$, and a single asterisk approaches significance at $0.5 < p < 0.10$. Placement of the asterisk(s) at the panel's time label denotes the relatively more tortuous species for that time interval, with grey asterisk(s) for Namibia animals and black asterisk(s) for Botswana animals.
(PDF)

**S14 Fig.** Frequency density of seasonal turning angles for (a) Etosha lions and (b) Chobe/Linyanti spotted hyenas during the nocturnal period according to various moon phases. Darkest lunar phase (far left panel) increasing to brightest lunar phase (far right panel).
(PDF)

**S15 Fig.** Tortuosity of (a) lions and (b) spotted hyenas from the Etosha National Park, Namibia (i & ii) and the Chobe National Park and Linyanti Conservancy, Botswana (iii & iv). Path tortuosity is depicted at different distance intervals from competitors (i & iii), and conspecifics (ii & iv).
(PDF)

**S16 Fig. Lion revisitation and duration (RD) space plots.** Time-use constructs of lion individuals from the (a) Etosha National Park, Namibia; (b) Chobe National Park; (c) Linyanti Conservancy; and (d) Okavango Delta, Botswana. Unique identifiers are depicted vertically on the left of each row of figures. α-LoCoH hulls of individual's utilization distributions (far left). Hull parent points colored by visitation rate (nsv, number of separate visits; second from left), and duration of visit (mnlv, mean number of locations in the hull per visit; third from left). RD space scatterplots (second from right) with X-axis = visitation rate (nsv), and Y-axis = duration of visit (mnlv), provide a legend for revisitation/duration (RD) values for the map (far right). Points in the RD space have been jiggled to better see point density, and each point represents a hull. Points on the maps are colored by their location in the RD space. Separate visits are defined by an inter-visit gap period $\geq 12$ hours. Hulls were created using the adaptive method. Duplicate points are offset by 1 map unit.
(PDF)

**S17 Fig. Spotted hyena revisitation and duration (RD) space plots.** Time-use constructs of spotted hyena individuals from the (a) Etosha National Park, Namibia; (b) Chobe National Park; and (c) Linyanti Conservancy, Botswana. Unique identifiers are depicted vertically on the left of each row of figures. α-LoCoH hulls of individual's utilization distributions (far left). Hull parent points colored by visitation rate (nsv, number of separate visits; second from left), and duration of visit (mnlv, mean number of locations in the hull per visit; third from left). RD space scatterplots (second from right) with X-axis = visitation rate (nsv), and Y-axis = duration of visit (mnlv), provide a legend for revisitation/duration (RD) values for the map (far right). Points in the RD space have been jiggled to better see point density, and each point represents a hull. Points on the maps are colored by their location in the RD space. Separate visits are defined by an inter-visit gap period $\geq 12$hours. Hulls were created using the adaptive method. Duplicate points are offset by 1 map unit.
(PDF)

**S18 Fig. Cluster analyses of lion revisitation and duration.** Maps (left panels) depict the individual lion's relocations as four (top row), six (middle row), and eight (bottom row) clusters in the (a) Etosha National Park, Namibia; (b) Chobe National Park; (c) Linyanti Conservancy; and (d) Okavango Delta, Botswana. Unique identifiers are depicted on top corner of each page. Relocations are color-coded according to the clusters indicated by the range of revisitation (number of separate visits) and duration (mean number of locations per visit) values in RD space plots (shown in central panels). Clusters in the RD space were determined with the *k*-prototype algorithm and are based on ecogeographical variables attached to each relocation. The smaller plots (right panels) present the distribution and percent category of clusters for each of the ecogeographical variables selected from the factor analysis of mixed data (FAMD). (PDF)

**S19 Fig. Cluster analyses of spotted hyena revisitation and duration.** Maps (left panels) depict the individual spotted hyena's relocations as four (top row), six (middle row), and eight (bottom row) clusters in the (a) Etosha National Park, Namibia; (b) Chobe National Park; and (c) Linyanti Conservancy, Botswana. Unique identifiers are depicted on top corner of each page. Relocations are color-coded according to the clusters indicated by the range of revisitation (number of separate visits) and duration (mean number of locations per visit) values in RD space plots (shown in central panels). Clusters in the RD space were determined with the *k*-prototype algorithm and are based on ecogeographical variables attached to each relocation. The smaller plots (right panels) present the distribution and percent category of clusters for each of the ecogeographical variables selected from the factor analysis of mixed data (FAMD). (PDF)

**S20 Fig. Relationship plots of lion and spotted hyena activity in relation to temperature for each hour of the 24-hour cycle.** Linear regression results relating the mean activity of (a) lions and (b) spotted hyenas to temperature (˚C) over each hour of the 24-hour cycle. Each panel indicates the time interval with its' adjusted $R^2$ and p-value. The red line is the line of best fit to the data, with the grey shaded bars the 95% confidence interval. An asterisk denotes significance at the alpha level with $^* < 0.05$, and $^{***} < 0.001$. (PDF)

## Acknowledgments

We thank Christopher Diab, Danica Metlay, Jacqueline Moser, Julie Fabricius Faustrup, Astri Frafjord, Linda Molloy, Christopher Mastropietro for dedicated field work, and Shayne Kotting, Mathews Daniel, Fredrick Khumub, Malakia Hango, Isaskar Uahoo, Dr. Rob Jackson, Dr. Caron Botes, and Dr. Larry Patterson for support and assistance in the field. We also thank the Ministry of Environment and Tourism Namibia and Boas Erckie (Etosha National Park), the Department of Wildlife and National Parks Botswana and Dr. Michael Flyman (Chobe National Park), the Chobe Enclave Community Trust (Linyanti Conservancy), the Okavango Kopano Mokoro Community Trust and S.K. Moepedi (OKMCT), Sanctuary Retreats and Charl Badenhorst for infrastructure support and permissions. We are grateful to the Etosha Ecological Institute and CARACAL Biodiversity Center for the use of lab facilities, and to Dr. Richard Fynn and the Okavango Research Institute for materials. We thank Chris Coetzee and Fritz Stolzenberg in Namibia, and Trish Williams in Botswana for support outside the parks. We are indebted to Drs. Jerry and Jana Lackey for their constant and unwavering support. We are also grateful to Lotek Wireless Inc., Shush Productions, African Lion and Environmental Research Trust (ALERT), the Namibian Environment & Wildlife Society (NEWS), Gert Uls and Resun GmbH for sponsoring some of the field equipment. We thank Helicopter Horizons

for their assistance with animal captures, and to Dr. John 'Tico' McNutt and Mike Holding for their assistance with flight-tracking. We also thank Andy Lyons for assistance with T-LoCoH, Dana Seidel for guidance with coding R analyses, and Dr. Wolfgang Beyer for undertaking serological work.

## Author Contributions

**Conceptualization:** Nancy A. Barker, Rob Slotow, Wayne M. Getz.

**Data curation:** Nancy A. Barker, Vincent Stowbunenko.

**Formal analysis:** Nancy A. Barker, Vincent Stowbunenko.

**Funding acquisition:** Nancy A. Barker, Rob Slotow, Wayne M. Getz.

**Investigation:** Nancy A. Barker.

**Methodology:** Nancy A. Barker, Wayne M. Getz.

**Project administration:** Nancy A. Barker.

**Resources:** Nancy A. Barker, Francois G. Joubert, Marthin Kasaona, Gabriel Shatumbu, Kathleen A. Alexander, Rob Slotow, Wayne M. Getz.

**Software:** Nancy A. Barker, Vincent Stowbunenko.

**Supervision:** Nancy A. Barker, Kathleen A. Alexander, Rob Slotow, Wayne M. Getz.

**Validation:** Nancy A. Barker.

**Visualization:** Nancy A. Barker.

**Writing – original draft:** Nancy A. Barker.

**Writing – review & editing:** Nancy A. Barker, Rob Slotow, Wayne M. Getz.

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
