## [Decision Letter · Decision Letter 0]

30 May 2022

PONE-D-22-05250Coursing hyenas and stalking lions: the potential for inter- and intraspecific interactionsPLOS ONE

Dear Dr.Baker

Thank you for submitting your manuscript to PLOS ONE. After careful consideration, we feel that it has merit but does not fully meet PLOS ONE’s publication criteria as it currently stands. Therefore, we invite you to submit a revised version of the manuscript that addresses the points raised during the review process.  The referee and I completely agree on the merit of your work and I think that after revising the text following the main advice to simplify presentation trough the use of tables your work will be suitable for publication.

We look forward to receiving your revised manuscript.

Kind regards,

Marco Apollonio

Academic Editor

PLOS ONE

Journal Requirements:

3. We note that you have referenced (ie. Bewick et al. [5]) which has currently not yet been accepted for publication. Please remove this from your References and amend this to state in the body of your manuscript: (ie “Bewick et al. [Unpublished]”) as detailed online in our guide for authors

5. Please include your tables as part of your main manuscript and remove the individual files. Please note that supplementary tables (should remain/ be uploaded) as separate "supporting information" files.

Reviewers' comments:

Reviewer's Responses to Questions

**Comments to the Author**

1. Is the manuscript technically sound, and do the data support the conclusions?

Reviewer #1: Yes

2. Has the statistical analysis been performed appropriately and rigorously? 

Reviewer #1: Yes

3. Have the authors made all data underlying the findings in their manuscript fully available?

Reviewer #1: Yes

4. Is the manuscript presented in an intelligible fashion and written in standard English?

Reviewer #1: Yes

5. Review Comments to the Author

Reviewer #1: I really liked your paper and feel it presents really important information from a great dataset and hence recommended minor revisions. That said, it was a pretty tough read - very long and filled with data that I think would be better served being presented as tables or figures (I realise there are a lot of these already - but they help your audience out a lot). I think this would help with the flow of the manuscript also. Hence, my big suggestion would be to try to move some of the Results text into tables, and abbreviate the manuscript wherever you can.

Specific suggestions:

- L61: I'd argue that these species 'preferentially' forage on different types of food, but opportunism masks these trends to a degree (Hayward, M. W. and Kerley, G. I. H. 2008. Prey preferences and dietary overlap amongst Africa's large predators. - SA J. Wildl. Res. 38: 93-108.).

- L326: Can you add in the years that these data were collected here again please?

- L498: Somewhere around here, I'd be interested in hearing about how the core areas were situated in relation to those of their competitor (i.e., distances from centre of core areas or close to territory boundaries)?

- L555: Change to "Moreover, lion and hyena movements and activity differed according to the..."

- LL596-608: Can you relate movements to temperatures as well (each hour of the day)? Seems you've tested one potential explanation for activity, but ignored this one that Rob and I used to explain the strict nocturnality of hyaenas.

- L603: Is this clearer - "Despite lions and hyaenas exhibiting higher proportions of activity during certain periods of the night..."

- L793 and stats throughout: always present the chi-square value, alongside degrees of freedom and the p-value.

- L832: "As mainly cursorial predators, spotted hyaenas ..."

- L834-836: Isn't this a chicken and egg issue? Are they using those habitats because they like those habitats or their prey likes those habitats?

- L838: isn't preferred prey an important factor here alongside habitat?

- L844-846: Kasim Rafiq found something similar with leopards and the African carnivore guild as well.

I hope that is helpful.

Sincerely

Matt Hayward

6. PLOS authors have the option to publish the peer review history of their article (what does this mean?). If published, this will include your full peer review and any attached files.

Reviewer #1: **Yes: **Matt Hayward

---

## [Author Response · Author response to Decision Letter 0]

28 Dec 2022

1. We have redone the file naming system and corrected this to ensure that our manuscript now meets PLOS ONE’s style requirements for file naming.

2. We provided in our submission the repository information for which our data is held in, within the Zenodo repository and can be found here: Zenodo: https://doi.org/10.5281/zenodo.5949231

3. We do not find the reference Bewick et al. in our paper. Our reference [5] is for Scognamillo et al., and we have checked through our paper for any reference to Bewick and have found none.

4. We created Fig 1 ourselves using ArcGIS (ESRI ArcMap v.10.0). The previous PDF of Fig 1 uploaded to PLOS ONE was mistakenly created using the screenshot tool instead of exporting as PDF directly from ArcGIS. This resulted in lines around the map, which made it look as if had been pasted from somewhere else. We have since corrected this error by exporting directly from ArcGIS as PDF.

5. Table 1 has now been included into the main manuscript, and the individual files have been removed.

6. We have reviewed our reference list to ensure that it is complete and correct. 

Reviewer #1: 

We have created new tables (Tables 3, 4, and 7) and included significant values (into Tables 5, 6, and 8) taken from supplementary materials to replace approximately 4 pages of text in the Results section. 

Specific suggestions:

- L61: 

We have changed “More specifically, species whose ranges overlap forage different types of food” to “More specifically, species whose ranges overlap forage different types of food, either preferentially or opportunistically” (L64) to reflect this, and included the reference to Hayward and Kerley.

- L326: 

We have added in the years that the data were collected here (L343-351).

- L498: 

We included this information at the end of this section (L522-523) and with a new table in supplementary materials (S6 Table) in where we said:

“Furthermore, individual core areas were more distant from competitor core areas than to the nearest territory boundaries of competitors’. This was significant for most cases (all t-tests p < 0.05), except for hyenas to heterospecific competitors in Etosha, and to conspecific competitors in Botswana.” and referred to the new table in the supplementary materials. 

- L555: 

We changed “Moreover, lion and hyena movements were found to differ according to the …” to “Moreover, lion and hyena movements and activity differed according to the …” (L585) as recommended.

- L596-608: 

We have added in L556-560 a section indicating the relationship between lion and hyena activity to temperature, in which we also provided the adjusted R2 and included this information in a new figure in S20 Fig in supplementary materials. 

- L603: 

We changed “Despite whether lions and hyenas were exhibiting higher proportions of activity during certain periods of the night” to “Despite lions and hyenas exhibiting higher proportions of activity during certain periods of the night” (L652) as recommended.

- L793 and stats throughout: 

We have presented the chi-square values alongside with the degrees of freedom and p-values throughout the manuscript where mentioned (i.e., L666, L668, L669), including the tables and in supplementary materials.

- L832: 

We changed “Mainly cursorial predators, spotted hyenas …” to “As mainly cursorial predators, spotted hyenas” (L865) as recommended.

- L834-836: 

We state here that lions and spotted hyenas within this study were utilizing similar habitats, not whether they or their prey likes those habitats, and referenced this to the study of Durant et al., 2010 which found that lions and spotted hyenas exhibited broad overlap in terms of habitat preferences, regardless of prey (L867-869).

- L838: 

We agree prey is an important factor for the distribution and movements of carnivores, and we include this later in our discussion L888-899 where we state: 

“Previous studies have demonstrated lion movements to be inextricably tied to the location of water-holes across the landscape due to concentrated search efforts for prey [25,118]. …” and later “Spotted hyenas have been found to use locations far from water for den sites and resting [30,114], thus we suggest that hyenas are potentially choosing to remain in areas with landscape characteristics that minimizes detection [120] while increasing prey vulnerability [121].” 

In the earlier section of the discussion, we focus here on the differences in the use of similar habitats between the two species, regardless of prey. 

- L844-846: 

We have included this recent reference here (L877-879).

---

## [Editor Report · Decision Letter 1]

17 Jan 2023

Coursing hyenas and stalking lions: the potential for inter- and intraspecific interactions

PONE-D-22-05250R1

Dear Dr. Baker

We’re pleased to inform you that your manuscript has been judged scientifically suitable for publication and will be formally accepted for publication once it meets all outstanding technical requirements.

Kind regards,

Marco Apollonio

Academic Editor

PLOS ONE